# Derivation of trophoblast stem cells from naïve human pluripotent stem cells

**Chen Dong[1,2], Mariana Beltcheva[1,2], Paul Gontarz[1,2], Bo Zhang[1,2], Pooja Popli[3], Laura A Fischer[1,2], Shafqat A Khan[1,2], Kyoung-mi Park[1,2], Eun-Ja Yoon[1,2], Xiaoyun Xing[2,4], Ramakrishna Kommagani[3], Ting Wang[2,4], Lilianna Solnica-Krezel[1,2], Thorold W Theunissen[1,2]***

[1]Department of Developmental Biology, Washington University School of Medicine, St. Louis, United States; [2]Center of Regenerative Medicine, Washington University School of Medicine, St. Louis, United States; [3]Department of Obstetrics and Gynecology, Center for Reproductive Health Sciences, Washington University School of Medicine, St. Louis, United States; [4]Department of Genetics, Center for Genome Sciences & Systems Biology, Washington University School of Medicine, St. Louis, United States

**Abstract** Naïve human pluripotent stem cells (hPSCs) provide a unique experimental platform of cell fate decisions during pre-implantation development, but their lineage potential remains incompletely characterized. As naïve hPSCs share transcriptional and epigenomic signatures with trophoblast cells, it has been proposed that the naïve state may have enhanced predisposition for differentiation along this extraembryonic lineage. Here we examined the trophoblast potential of isogenic naïve and primed hPSCs. We found that naïve hPSCs can directly give rise to human trophoblast stem cells (hTSCs) and undergo further differentiation into both extravillous and syncytiotrophoblast. In contrast, primed hPSCs do not support hTSC derivation, but give rise to non-self-renewing cytotrophoblasts in response to BMP4. Global transcriptome and chromatin accessibility analyses indicate that hTSCs derived from naïve hPSCs are similar to blastocyst-derived hTSCs and acquire features of post-implantation trophectoderm. The derivation of hTSCs from naïve hPSCs will enable elucidation of early mechanisms that govern normal human trophoblast development and associated pathologies.

*For correspondence:
t.theunissen@wustl.edu

## Introduction

Mammalian pluripotency spans a continuum of discrete but interconvertible states, each with a distinct set of molecular and functional attributes. Prior to implantation, the pluripotent epiblast compartment within the inner cell mass (ICM) of the blastocyst constitutes a naïve or ground state of pluripotency (*Nakamura et al., 2016*; *Nichols and Smith, 2009*; *Stirparo et al., 2018*). This naïve state can be captured in vitro in the form of mouse embryonic stem cells (mESCs). After implantation, transcription factors associated with naïve pluripotency are downregulated and pluripotent cells become primed for differentiation in response to signals from the surrounding extraembryonic tissues. Human embryonic stem cells (hESCs) derived under conventional conditions are thought to represent a primed pluripotent state, and were shown to correspond transcriptionally to the late post-implantation epiblast in a non-human primate model (*Nakamura et al., 2016*; *Nichols and Smith, 2009*). Much effort has been made in recent years to develop strategies for capturing hESCs in a naïve pluripotent state (*Chan et al., 2013*; *Gafni et al., 2013*; *Hanna et al., 2010*; *Qin et al., 2016*; *Takashima et al., 2014*; *Theunissen et al., 2014*; *Ware et al., 2014*; *Zimmerlin et al., 2016*). In particular, two transgene-free culture systems, 5i/L/A and t2i/L/Gö, were shown to induce defining transcriptional and epigenetic features of the human pre-implantation epiblast (*Huang et al., 2014*;

**eLife digest** The placenta is one of the most important human organs, but it is perhaps the least understood. The first decision the earliest human cells have to make, shortly after the egg is fertilized by a sperm, is whether to become part of the embryo or part of the placenta. This choice happens before a pregnancy even implants into the uterus. The cells that commit to becoming the embryo transform into 'naïve pluripotent' cells, capable of becoming any cell in the body. Those that commit to becoming the placenta transform into 'trophectoderm' cells, capable of becoming the two types of cell in the placenta. Placental cells either invade into the uterus to anchor the placenta or produce hormones to support the pregnancy.

Once a pregnancy implants into the uterus, the naïve pluripotent cells in the embryo become 'primed'. This prevents them from becoming cells of the placenta, and it poses a problem for placental research. In 2018, scientists in Japan reported conditions for growing trophectoderm cells in the laboratory, where they are known as "trophoblast stem cells". These cells were capable of transforming into specialized placental cells, but needed first to be isolated from the human embryo or placenta itself.

Dong et al. now show how to reprogram other pluripotent cells grown in the laboratory to produce trophoblast stem cells. The first step was to reset primed pluripotent cells to put them back into a naïve state. Then, Dong et al. exposed the cells to the same concoction of nutrients and chemicals used in the 2018 study. This fluid triggered a transformation in the naïve pluripotent cells; they started to look like trophoblast stem cells, and they switched on genes normally active in trophectoderm cells. To test whether these cells had the same properties as trophoblast stem cells, Dong et al. gave them chemical signals to see if they could mature into placental cells. The stem cells were able to transform into both types of placental cell, either invading through a three-dimensional gel that mimics the wall of the uterus or making pregnancy hormones.

There is a real need for a renewable supply of placental cells in pregnancy research. Animal placentas are not the same as human ones, so it is not possible to learn everything about human pregnancy from animal models. A renewable supply of trophoblast stem cells could aid in studying how the placenta forms and why this process sometimes goes wrong. This could help researchers to better understand miscarriage, pre-eclampsia and other conditions that affect the growth of an unborn baby. In the future, it may even be possible to make custom trophoblast stem cells to study the specific fertility issues of an individual.

---

*Liu et al., 2017*; *Stirparo et al., 2018*; *Takashima et al., 2014*; *Theunissen et al., 2016*; *Theunissen et al., 2014*).

The isolation of naïve hESCs provides a cellular experimental platform to interrogate aspects of human pre-implantation development that are difficult to study in primed hESCs. For example, naïve hESCs have offered insights into the mechanisms governing X-linked dosage compensation (*Sahakyan et al., 2017*), the role of human-specific transposable elements that are expressed in the pre-implantation embryo (*Pontis et al., 2019*; *Theunissen et al., 2016*), and the mechanisms leading to activation of naïve-specific enhancers (*Pastor et al., 2018*). Naïve hESCs may also afford a platform for dissecting cell fate decisions in the early human embryo (*Dong et al., 2019*). While naïve hESCs are unresponsive to direct application of embryonic inductive cues, they acquire the capacity to undergo efficient multi-lineage differentiation upon treatment with a Wnt inhibitor, a process called 'capacitation' (*Rostovskaya et al., 2019*). This process is thought to reflect the requirement for dismantling of the naïve transcriptional program upon implantation in vivo, and the acquisition of a differentiation-competent formative phase (*Smith, 2017*).

Molecular profiling of naïve hESCs has suggested that these cells may harbor a predisposition towards human extraembryonic cell fates. Gene expression studies revealed a pronounced upregulation in naïve relative to primed hESCs of trophoblast-associated transcription factors, including *ELF3, GCM1,* and *TFAP2C* (*Theunissen et al., 2016*). In addition, chromatin accessibility studies indicated that naïve hESCs share a broad panel of open chromatin sites with first-trimester placental tissues (*Pontis et al., 2019*). Intriguingly, embryonic and extraembryonic lineage markers are briefly co-expressed in the late morula and early blastocyst according to single cell RNA-seq (scRNA-seq)

studies of human pre-implantation embryos (*Petropoulos et al., 2016*). This is precisely the stage of human development that displays the closest correspondence to naïve hESCs based on the expression patterns of transposable elements (*Theunissen et al., 2016*). Thus, we surmised that current methodologies for inducing naïve human pluripotency may yield a pre-implantation identity that is competent for both embryonic and extraembryonic differentiation. Here, using three independent methodologies, we find that naïve hPSCs have enhanced capacity for differentiation along the trophoblast lineage relative to primed hPSCs. In particular, we show that when cultured in human trophoblast stem cell (hTSC) media (*Okae et al., 2018*), naïve hPSCs can directly give rise to hTSCs, as confirmed by morphological, molecular, and transcriptomic criteria. We have also profiled the chromatin accessibility landscape of hTSCs for the first time, thus providing a valuable resource to identify potential regulatory elements and transcriptional determinants of human trophoblast development.

## Results

### Naïve hESCs exhibit increased trophoblast potential during embryoid body formation

As a first step toward examining the trophoblast potential of naïve and primed hESCs, we measured the expression levels of trophoblast-associated markers during embryoid body (EB) formation (*Figure 1A*), which provides a rapid assessment of spontaneous differentiation capacity into early lineages (*Allison et al., 2018*). Previous studies reported limited induction of embryonic lineage markers in EBs formed from naïve hESCs, but did not examine the expression of trophoblast-associated genes (*Liu et al., 2017*; *Rostovskaya et al., 2019*). We generated naïve hESCs in 5i/L/A (*Theunissen et al., 2014*) from two genetic backgrounds, H9 and WIBR3, confirmed their upregulation of naïve-specific markers and downregulation of primed-specific markers (*Figure 1—figure supplement 1A*), and aggregated them to form EBs in growth factor- and inhibitor-free media for 12 days (*Figure 1A*; *Figure 1—figure supplement 1B*). The mRNA expression levels of six trophoblast markers, *ELF5*, *KRT7*, *TFAP2C*, *GATA3*, *TEAD4*, and *CDX2* (*Hemberger et al., 2010*; *Lee et al., 2016*; *Ng et al., 2008*; *Strumpf et al., 2005*), were measured by quantitative real time PCR (qRT-PCR) analysis (*Figure 1B*; *Figure 1—figure supplement 1C*). Significantly enriched expression of trophoblast markers was observed in EBs derived from naïve compared to primed hESCs (*Figure 1B*; *Figure 1—figure supplement 1C*). Furthermore, many of the examined trophoblast markers were already elevated in naïve versus primed hESCs prior to EB formation (*Figure 1B*; *Figure 1—figure supplement 1C*), consistent with our prior transcriptome analysis (*Theunissen et al., 2016*). These findings support the notion that naïve hESCs harbor increased spontaneous trophoblast differentiation potential compared to primed hESCs.

### Naïve hESCs require capacitation or re-priming to respond to BMP4-directed trophoblast differentiation

We next assessed the trophoblast potential of naïve relative to primed hESCs using a protocol for directed trophoblast differentiation that utilizes a low dose of bone morphogenetic protein 4 (BMP4) (*Horii et al., 2016*). It has long been known that primed hPSCs acquire certain trophoblast characteristics upon stimulation with BMP4 (*Amita et al., 2013*; *Horii et al., 2016*; *Xu et al., 2002*). However, when naïve hESCs were subjected to a protocol for BMP4-directed differentiation into cytotrophoblast (CTB) progenitors (*Horii et al., 2016*), the cells did not survive (*Figure 1C*). This recalcitrance to BMP4 is reminiscent of the delayed response of naïve hPSCs to embryonic inductive cues (*Liu et al., 2017*; *Rostovskaya et al., 2019*). We examined whether naïve hESCs would gain the capacity for BMP4-directed differentiation upon returning to the primed state, a process referred to as 're-priming' (*Theunissen et al., 2016*). Indeed, naïve hESCs treated with StemPro for five passages re-gained competence for BMP4-directed trophoblast differentiation, as shown by the expression of several CTB markers (*Figure 1D,E*; *Figure 1—figure supplement 2A,B*). When these CTBs were further differentiated using feeder-conditioned medium supplemented with BMP4, we observed the expected induction of extravillous trophoblast (EVT) and syncytiotrophoblast (STB) marker genes, as observed for differentiation from primed hESCs (*Horii et al., 2016*; *Figure 1G*; *Figure 1—figure supplements 1D* and *2A,B*). We also examined whether capacitation of naïve hESCs

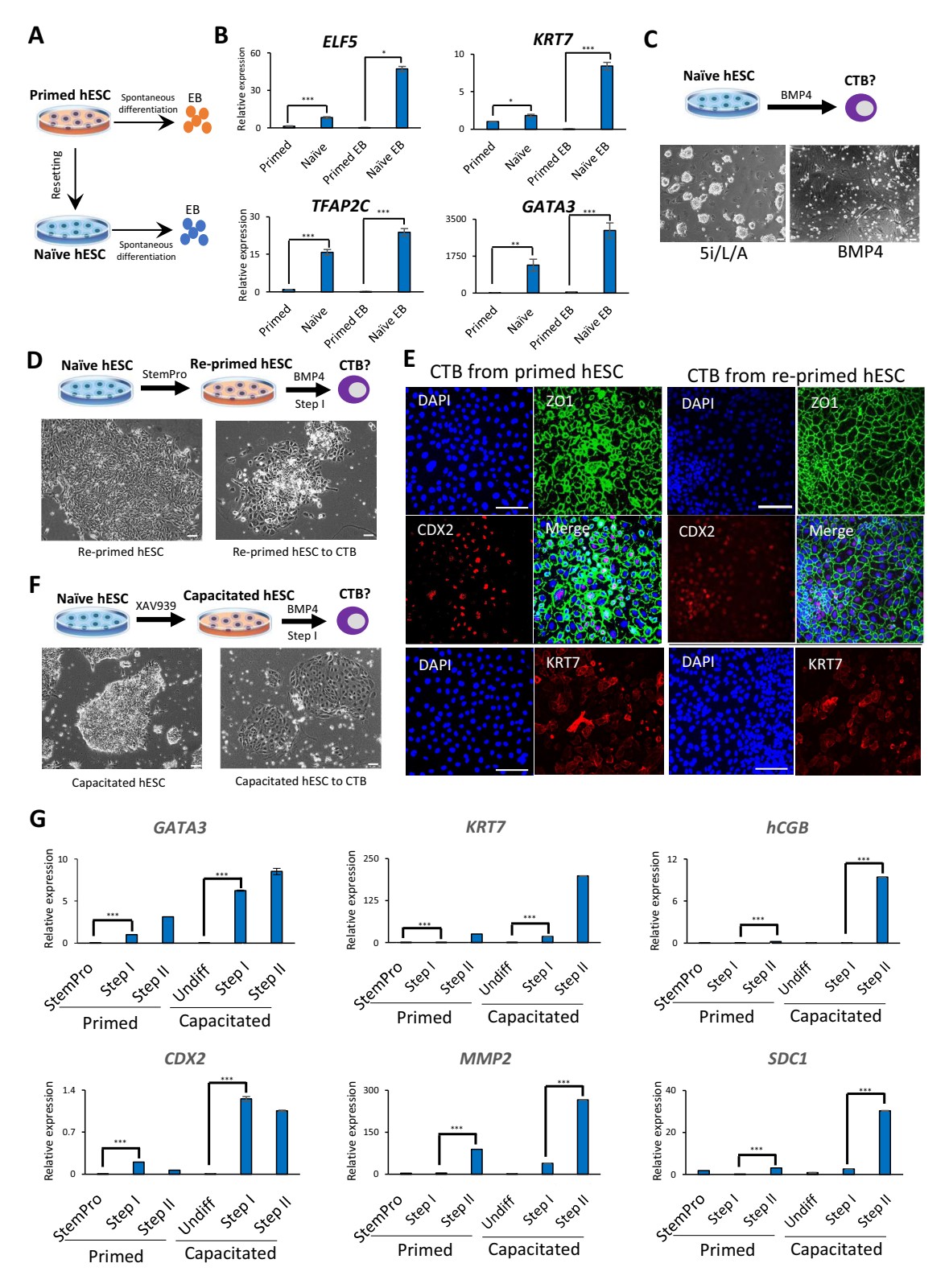

**Figure 1.** Trophoblast potential of different hPSC states under spontaneous and BMP4-mediated differentiation conditions. (**A**) The experimental scheme for assessing spontaneous trophoblast differentiation potential of primed and naïve hPSCs by using EB formation assay. (**B**) Quantitative gene expression analysis for trophoblast marker genes *ELF5, KRT7, TFAP2C,* and *GATA3* in H9 primed and naïve hPSCs and EBs generated from the respective hPSCs. Error bars indicate ±1 SD of technical replicates. '*' indicates a p-value<0.05, '**' indicates a p-value<0.01, and '***' indicates a

*Figure 1 continued on next page*

*Figure 1 continued*

p-value<0.001. (C) The experimental scheme for assessing BMP4-mediated trophoblast differentiation potential of naive hESCs (top). Phase contrast image of H9 naïve hESCs and H9 naïve hESCs following the 6 day step I protocol for BMP4-mediated CTB differentiation (bottom). The scale bars indicate 75 μm. (D) The experimental scheme of re-priming naïve hESCs and assessing their BMP4-mediated trophoblast differentiation potential (top). Phase contrast images of H9 re-primed hESCs and H9 re-primed hESCs following the 6 day step I protocol for BMP4-mediated CTB differentiation (bottom). The scale bars indicate 75 μm. (E) Immunofluorescence staining for CDX2, ZO1, and KRT7 in CTBs generated from H9 primed and re-primed hESCs. The scale bars indicate 75 μm. (F) The experimental scheme of capacitating naïve hESCs and assessing their BMP4-mediated trophoblast differentiation potential (top). Phase contrast images of H9 capacitated hESCs and H9 capacitated hESCs following the 6 day step I protocol for BMP4-mediated CTB differentiation (bottom). The scale bars indicate 75 μm. (G) Quantitative gene expression analysis for trophoblast marker genes *GATA3, CDX2, KRT7, MMP2, hCGB*, and *SDC1* in H9 primed and capacitated hESCs and trophoblasts differentiated from the respective hPSCs. Error bars indicate ± 1 SD of technical replicates. '***' indicates a p-value<0.001.

The online version of this article includes the following figure supplement(s) for figure 1:

**Figure supplement 1.** Trophoblast potential of different hPSC states under spontaneous and BMP4-mediated differentiation conditions.
**Figure supplement 2.** Immunofluorescence staining of trophoblast markers.

using the Wnt inhibitor, XAV939 (*Rostovskaya et al., 2019*), would confer responsiveness to BMP4-directed trophoblast differentiation. Capacitated cells could not only readily undergo Step I CTB differentiation, but did so more efficiently than primed hESCs based on analysis of trophoblast-specific transcripts and proteins (*Figure 1F,G*; *Figure 1—figure supplements 1D* and *2C*). These results indicate that an efficient response to BMP4-directed trophoblast differentiation requires exit from naïve human pluripotency, and appears to be most efficient when initiated from the formative phase.

## Human trophoblast stem cells can be derived from naïve, but not primed, hPSCs

The above experiments indicate that naïve hESCs efficiently upregulate trophoblast markers during spontaneous EB differentiation, but require transition into a formative pluripotent state to become competent for BMP4-directed trophoblast differentiation. This led us to consider whether naïve hESCs might be directly responsive to alternative conditions for trophoblast differentiation that are independent of BMP4. A recent study reported the derivation of hTSCs from blastocysts and first-trimester placental tissues in the presence of recombinant epidermal growth factor (EGF) and Wnt activator (CHIR), and inhibitors of transforming growth factor beta (TGFβ), histone deacetylase (HDAC), and Rho-associated kinase (ROCK) (*Okae et al., 2018*). We seeded isogenic lines of naïve and primed hPSCs on Collagen IV and examined their response to hTSC media (*Figure 2A*). Naïve hPSCs acquired a typical hTSC-like morphology within several passages and could be expanded for at least 20 passages while maintaining a high proliferation rate (*Figure 2B*; *Figure 2—figure supplement 1A*). Similar results were obtained using three independent naïve hPSC lines derived from H9 hESCs, WIBR3 hESCs, and AN1 induced pluripotent stem cells (iPSCs). In contrast, the parental primed hPSCs did not acquire an hTSC-like morphology, even after prolonged culture in hTSC media (*Figure 2B*). These observations indicate that naïve hPSCs are capable of adapting to the specific culture environment of hTSCs, whereas primed hPSCs are not.

We proceeded to further characterize the naïve hPSC-derived hTSC-like cells (from here referred to as naïve hTSCs). We found that naïve hTSCs uniformly express ITGA6 and EGFR, two commonly used cell surface markers that mark both CTBs and hTSCs (*Bischof and Irminger-Finger, 2005*; *Horii et al., 2016*; *Okae et al., 2018*), in contrast to primed hPSCs that had been expanded in hTSC media (*Figure 2C*; *Figure 2—figure supplement 1B*). The naïve hTSCs also expressed significantly higher levels of *ELF5, KRT7, GATA3, TFAP2C*, and *TEAD4* transcripts (*Figure 2D*; *Figure 2—figure supplement 1C*). Notably, naïve hTSCs exhibited almost no *CDX2* expression, which is consistent with hTSCs derived from blastocysts or first-trimester placental tissues (*Okae et al., 2018*; *Figure 2D*; *Figure 2—figure supplement 1C*). Naïve hTSCs also expressed the hTSC markers KRT7, TEAD4, and TP63 at the protein level (*Lee et al., 2016*; *Li et al., 2013*; *Figure 2E*; *Figure 2—figure supplement 1D*). Flow cytometry analysis for CD75 and SUSD2, two naïve-specific cell surface markers (*Bredenkamp et al., 2019a*; *Collier et al., 2017*), confirmed the loss of naïve identity in naïve hTSCs (*Figure 2F*), which was further corroborated by downregulation of *KLF17* transcript (*Figure 2G*; *Figure 2—figure supplement 1E*). These findings indicate that naïve hPSCs directly

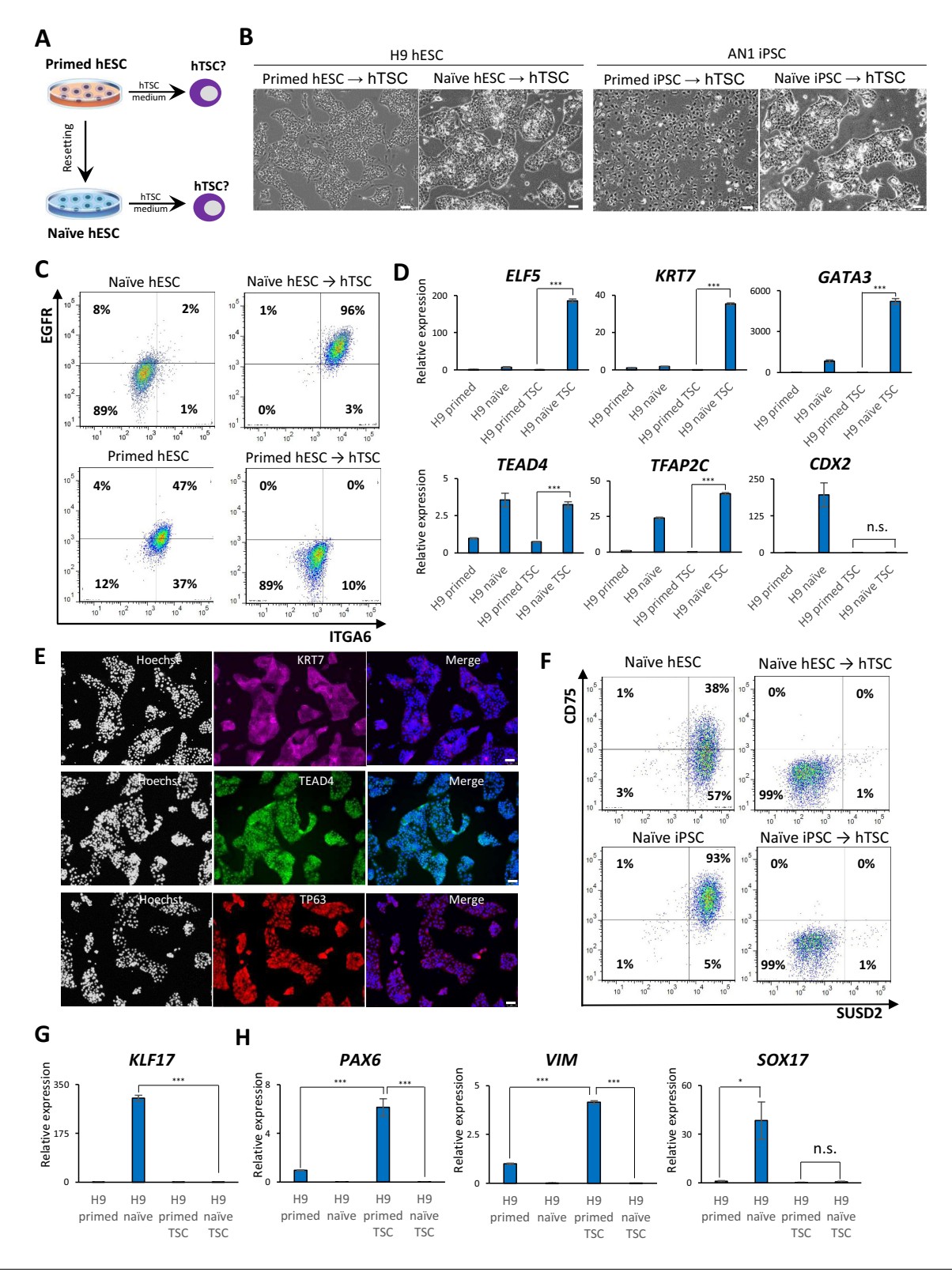

**Figure 2.** Examining the response of naïve and primed hPSCs to conditions for hTSC derivation. (A) The experimental scheme for deriving hTSCs from primed and naïve hPSCs. (B) Phase contrast images of H9 and AN_1.1 hTSC-like cells derived from naïve hPSCs, as well as H9 and AN_1.1 primed hPSCs following culture in hTSC medium. All H9 cells were at passage 8, and all AN_1.1 cells were at passage 10. The scale bars indicate 75 μm. (C) Flow cytometry analysis for TSC markers ITGA6 and EGFR in H9 naïve hPSCs, H9 hTSC-like cells derived from naïve hPSCs, H9 primed hPSCs, and H9

*Figure 2 continued on next page*

*Figure 2 continued*

primed hPSCs cultured in hTSC medium. (**D**) Quantitative gene expression analysis for trophoblast marker genes *ELF5, KRT7, GATA3, TFAP2C, TEAD4,* and *CDX2* in H9 primed and naïve hPSCs, H9 hTSC-like cells derived from naïve hPSCs, and H9 primed hPSCs cultured in hTSC medium. Error bars indicate ±1 SD of technical replicates. '***' indicates a p-value<0.001. (**E**) Immunofluorescence staining for TSC markers KRT7, TEAD4, and TP63 in H9 hTSC-like cells derived from naïve hPSCs. The scale bars indicate 75 µm. (**F**) Flow cytometry analysis for naïve hPSC markers SUSD2 and CD75 in H9 and AN_1.1 naïve hPSCs and hTSC-like cells derived from naïve hPSCs. (**G**) Quantitative gene expression analysis for naïve hPSC marker *KLF17* in H9 primed and naïve hPSCs, hTSC-like cells derived from naïve hPSCs, and primed hPSCs cultured in hTSC medium. Error bars indicate ±1 SD of technical replicates. '***' indicates a p-value<0.001. (**H**) Quantitative gene expression analysis for the embryonic germ layer markers *PAX6, VIM,* and *SOX17* in H9 primed and naïve hPSCs, hTSC-like cells derived from naïve hPSCs, and primed hPSCs cultured in hTSC medium. Error bars indicate ± 1 SD of technical replicates. '*' indicates a p-value<0.05, '***' indicates a p-value<0.001, and 'n.s.' indicates a p-value>0.05.

The online version of this article includes the following figure supplement(s) for figure 2:

**Figure supplement 1.** Derivation of hTSCs from naïve hPSCs.
**Figure supplement 2.** Derivation of naïve hTSCs from clonally expanded naïve hPSCs.

give rise to a uniform population of cells that closely resemble hTSCs based on gross morphology, surface markers, and trophoblast-specific gene expression. Conversely, primed hPSCs did not upregulate trophoblast markers when cultured in hTSC media, but instead exhibited increased expression of *VIM* and *PAX6*, suggesting the acquisition of a neuroectodermal fate (*Figure 2H*; *Figure 2—figure supplement 1F*).

To confirm that the ability of naïve hPSCs to give rise to hTSC-like cells is not specific to the 5i/L/A culture condition, we attempted to derive naïve hTSCs from naïve hPSCs cultured in an alternative naïve medium, PXGL (*Bredenkamp et al., 2019b*; *Figure 2—figure supplement 1G*). Both morphological and molecular analyses indicated that naïve hTSCs derived from PXGL-cultured naïve hPSCs closely resemble those derived from 5i/L/A-cultured naïve hPSCs (*Figure 2—figure supplement 1G–J*). This suggests that naïve hTSC-derivation is likely an intrinsic property of the naïve state of human pluripotency, regardless of the specific culture condition used. Additionally, we investigated whether naïve hPSCs can give rise to naïve hTSCs on a clonal level, which would preclude that hTSCs arise from a small population of pre-existing trophoblast-like cells in the culture. We therefore picked and expanded three single-cell H9 naïve hPSC clones (*Figure 2—figure supplement 2A*), and confirmed their naïve molecular characteristics and lack of hTSC marker expression (*Figure 2—figure supplement 2B,C*). Subsequently, naïve hTSCs were derived from these clonally expanded naïve hPSCs. All of these clonally derived naïve hTSCs exhibit a typical hTSC morphology (*Figure 2—figure supplement 2A*), and are negative for naïve hPSC markers but positive for hTSC markers (*Figure 2—figure supplement 2B,C*). These results lend further support to the notion that the ability to give rise to hTSCs is an intrinsic property of naïve hPSCs.

## hTSCs derived from naïve hPSCs have bipotent differentiation potential

Since naïve hTSCs exhibit numerous characteristics of hTSCs derived from blastocysts or placental tissues, we sought to examine their differentiation potential into specialized trophoblast cells. First, we performed directed differentiation of naïve hTSCs into EVTs, which invade the endometrium to increase blood flow between the mother and fetus (*James et al., 2012*; *Watson and Cross, 2005*). Following application of a culture system for EVT differentiation from hTSCs that contains Neuregulin 1 (NRG1), the TGF-β inhibitor A83-01, and Matrigel (*Okae et al., 2018*), naïve hTSCs acquired a characteristic EVT-like morphology (*Figure 3A,B*). We tested whether the naïve hTSC-derived EVT-like cells expressed two EVT-specific protein markers, HLA-G and MMP2 (*Horii et al., 2016*; *Lee et al., 2016*). Flow cytometry analysis demonstrated that about 90% of EVT-like cells were positive for HLA-G (*Figure 3C*). Similarly, expression and secretion of MMP2 was detected by ELISA and immunofluorescence staining (*Figure 3D*; *Figure 3—figure supplement 1A*). The induction of *HLA-G* and *MMP2* in naïve hTSC-derived EVT-like cells was also confirmed at the mRNA level by qRT-PCR (*Figure 3E*; *Figure 3—figure supplement 1B*). Finally, since invasiveness is a prominent property of EVTs (*Burton et al., 2009*; *James et al., 2012*; *McEwan et al., 2009*), we performed a transwell-based Matrigel invasion assay. The results indicate that our EVT-like cells have invasive potential, in contrast to the naïve hTSCs from which they were derived (*Figure 3F*; *Figure 3—figure supplement 1C*).

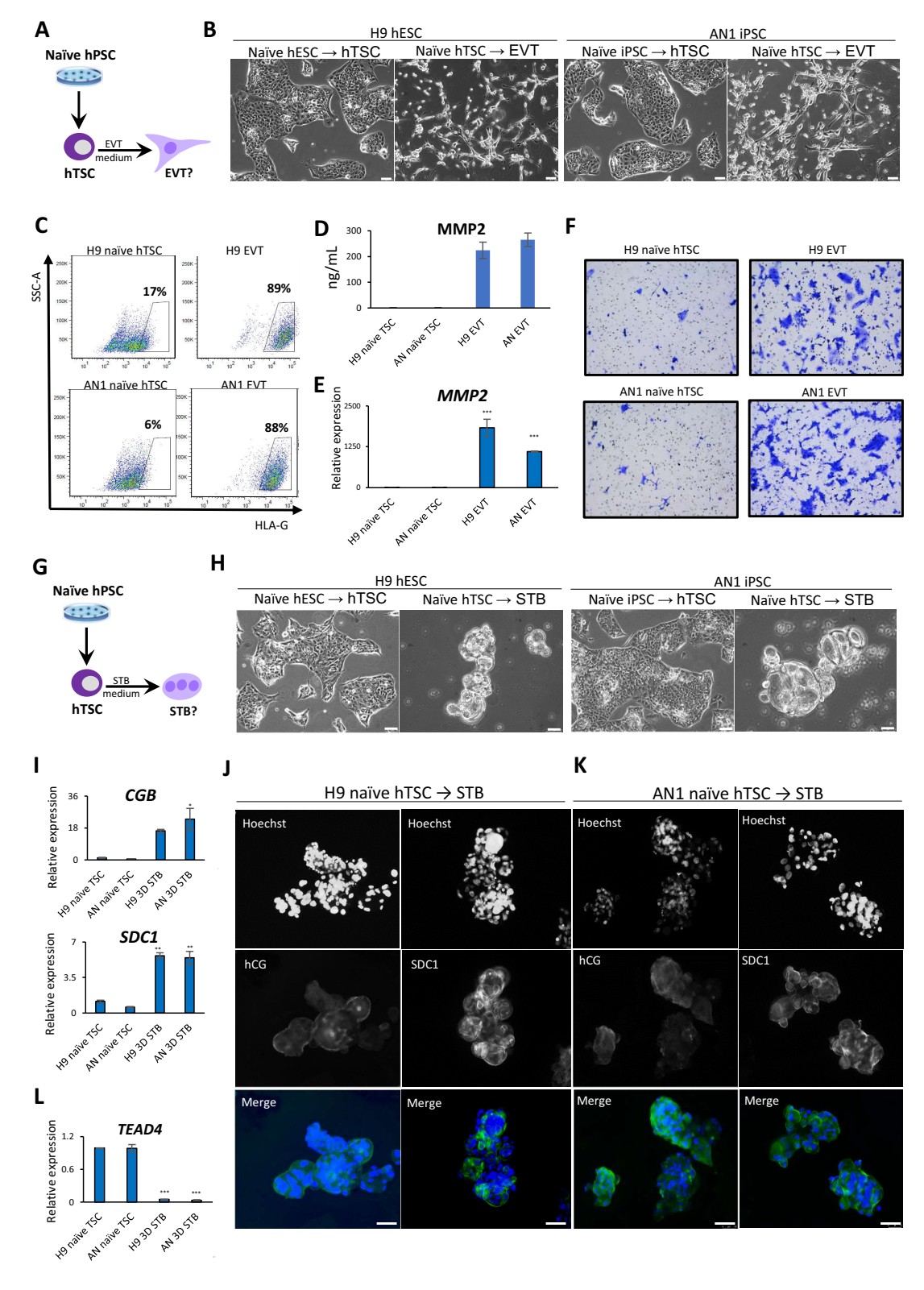

**Figure 3.** Directed differentiation of EVTs and STBs from naïve hPSC-derived hTSCs. (**A**) The experimental scheme for EVT differentiation from naïve hPSC-derived hTSCs. (**B**) Phase contrast image of H9 and AN_1.1 hTSC-like and EVT-like cells. The scale bars indicate 75 μm. (**C**) Flow cytometry analysis for EVT marker HLA-G in H9 and AN_1.1 hTSC-like cells and EVT-like cells. (**D**) Levels of MMP2 secreted by H9 and AN_1.1 hTSC-like cells and EVT-like cells as measured by ELISA. Error bars indicate ±1 SD of technical replicates. (**E**) Quantitative gene expression analysis for EVT marker gene

*Figure 3 continued on next page*

*Figure 3 continued*

MMP2 in H9 and AN_1.1 hTSC-like cells and EVT-like cells. Error bars indicate ± 1 SD of technical replicates. '***' indicates a p-value<0.001. (F) Matrigel invasion assay of H9 and AN_1.1 hTSC-like cells and EVT-like cells. (G) The experimental scheme for STB differentiation from naïve hPSC-derived hTSCs. (H) Phase contrast image of H9 and AN_1.1 hTSC-like and 3D STB-like cells. The scale bars indicate 75 μm. (I) Quantitative gene expression analysis for STB marker genes CGB and SDC1 in H9 and AN_1.1 hTSC-like cells and 3D STB-like cells. Error bars indicate ±1 SD of technical replicates. '*' indicates a p-value<0.05, and '**' indicates a p-value<0.01. (J) Immunofluorescence staining for STB markers hCG and SDC1 in H9 3D STB-like cells. The scale bars indicate 75 μm. (K) Immunofluorescence staining for STB markers hCG and SDC1 in AN_1.1 3D STB-like cells. The scale bars indicate 75 μm. (L) Quantitative gene expression analysis for TSC marker gene TEAD4 in H9 and AN_1.1 hTSC-like cells and 3D STB-like cells. Error bars indicate ± 1 SD of technical replicates. '***' indicates a p-value<0.001.

The online version of this article includes the following figure supplement(s) for figure 3:

**Figure supplement 1.** Directed differentiation of EVTs and STBs from naïve hPSC-derived hTSCs.

Second, we performed directed differentiation of naïve hTSCs into STBs, which are multinucleated cells that produce placental hormones and mediate maternal-fetal communication. We applied two protocols for differentiation of hTSCs into STBs involving either 2D or 3D culture in the presence of Forskolin (*Okae et al., 2018*; *Figure 3G*). The naïve hTSC-derived 2D STB-like cells showed characteristic morphological features, including multinucleation (*Figure 3—figure supplement 1D*). They also expressed the STB marker human chorionic gonadotropin (hCG) (*Figure 3—figure supplement 1E*). The naïve hTSC-derived 3D STB-like cells exhibited a cyst-like morphology typical for 3D STBs (*Okae et al., 2018*; *Figure 3H*), and expressed the STB markers hCG and SDC1 based on immunofluorescence and qRT-PCR analysis (*Jokimaa et al., 1998*; *Strumpf et al., 2005*; *Figure 3I–K*). Finally, qRT-PCR analysis confirmed downregulation of the hTSC-specific marker *TEAD4* (*Figure 3L*), indicating the loss of hTSC identity. These morphological and molecular data demonstrate that hTSCs derived from naïve hPSCs are capable, at least at the population level, of undergoing further differentiation into two specialized trophoblast cell types, EVT and STB.

## Global transcriptome and chromatin accessibility profiles of hTSCs derived from naïve hPSCs

To examine whether naïve hTSCs and their differentiated derivatives possess transcriptomic signatures of the trophoblast lineage, we sequenced total RNA isolated from three types of hPSCs (naïve, capacitated, and primed), blastocyst-derived BT5 hTSCs (*Okae et al., 2018*), naïve hTSCs and their differentiated progeny (EVT and STB), and primed hPSCs cultured in hTSC media. Principal component analysis (PCA) revealed that the samples clustered together based on cell type, rather than genetic background (*Figure 4A*). Pluripotent and trophoblast identities were predominantly divided along principal component 1 (PC1), which accounted for 41% of the variation in gene expression. In addition, naïve, capacitated, and primed hPSCs formed clearly defined clusters that were separated along PC2. It is worth noting that naïve hTSCs and BT5 hTSCs clustered together very closely on the PCA plot (*Figure 4A*). Further examination indicated that naïve hTSCs express key trophoblast marker genes at comparable or even higher levels than the *bona fide*, BT5 hTSCs (*Figure 4—figure supplement 1A*). Whereas naïve hTSCs were well-separated from the pluripotent samples, primed hPSCs acquired a very distinct transcriptional profile in hTSC media compared to naïve hTSCs. Naïve hTSCs displayed strong upregulation of many transcripts associated with trophoblast development, whereas primed cells instead acquired neuroectodermal characteristics in hTSC media (*Figure 4B* and *Supplementary file 1*). Hence, exposure to hTSC media induces a drastically different transcriptomic profile when applied to either naïve or primed hPSCs.

To further understand gene expression dynamics during naïve hTSC derivation and terminal EVT and STB differentiation, we identified six clusters of genes based on differential expression in naïve hPSCs, naïve hTSCs, and naïve hTSC-derived EVTs and STBs (*Figure 4C*). Cluster one includes genes that are enriched in naïve hPSCs, such as *DPPA5*, *KLF4*, *NANOG*, *POU5F1*, *SUSD2*, and *ZFP42* (*Figure 4D* and *Supplementary file 2*). This cluster is enriched in gene ontology (GO) terms associated with transcriptional regulation and embryonic development (*Supplementary file 3*). Cluster two includes genes such as *LRP5* and *TEAD4* that are enriched in both naïve hPSCs and naïve hTSCs

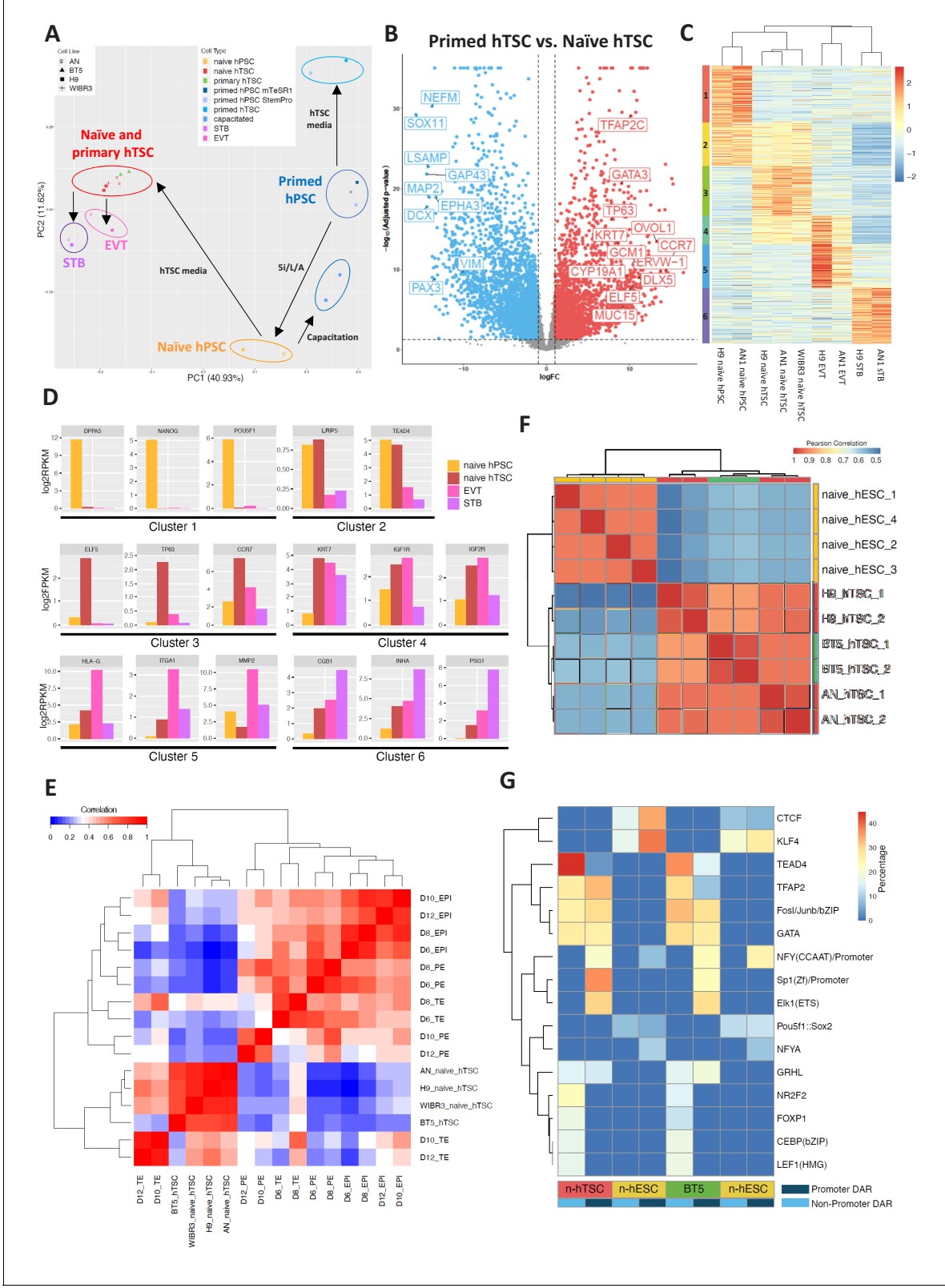

**Figure 4.** Transcriptomic and chromatin accessibility profiling of naïve hPSC-derived hTSCs. (**A**) Principal component analysis (PCA) of primed, capacitated, and naïve hPSCs, BT5 hTSCs, naïve hTSCs, EVTs, STBs, and primed hPSCs cultured in hTSC medium based on RNA-seq data. Circles were drawn around samples cultured in the same media. (**B**) Volcano plots showing fold change (X axis) between naïve hTSCs and primed hPSCs cultured in hTSC medium. The light blue dots represent genes that are the most significantly upregulated in primed hTSCs (defined as those that have a log fold

*Figure 4 continued*

change < −1 and wherein p<0.05). The red dots represent genes that are the most significantly upregulated in naïve hPSCs cultured in hTSC medium (defined as those that have a log fold change >1 and wherein p<0.05). (C) Heatmap of RNA-seq data from naïve hPSCs, naïve hTSCs, EVTs, and STBs (DEGs with more than 2X fold change and p-adj < 0.05 are analyzed). Cluster one represents genes highly expressed in naïve hPSCs only. Cluster two represents genes highly expressed in both naïve hPSCs and naïve hTSCs. Cluster three represents genes highly expressed in naïve hTSCs only. Cluster four represents genes highly expressed in both naïve hTSCs and EVTs. Cluster five represents genes highly expressed in EVTs only. Cluster six represents genes highly expressed in STBs only. (D) Expression levels of key marker genes associated with clusters 1–6. (E) Heatmap indicating the correlation among scRNA-seq data from published day 6, day 8, day 10, and day 12 TE, EPI, and PE (*Zhou et al., 2019*), as well as RNA-seq data from naïve and BT5 hTSCs. Genes specifically expressed in the TE, EPI, or PE lineages were analyzed (*Zhou et al., 2019*). (F) Heatmap of ATAC-seq data from naïve hESCs and naïve hTSCs (Pearson correlations were calculated based on genome-wise ATAC-seq signal). Naïve hESC ATAC-seq data were retrieved from a published dataset (*Pastor et al., 2018*). (G) Transcription factor binding motifs enriched in the promoter and non-promoter regions of the DARs specific to naïve hESCs vs. naïve hTSCs and naïve hESCs vs. BT5 hTSCs.

The online version of this article includes the following figure supplement(s) for figure 4:

**Figure supplement 1.** Transcriptomic profiling of naïve hPSC-derived hTSCs.
**Figure supplement 2.** Chromatin accessibility profiling of naïve hPSC-derived hTSCs.

(*Figure 4D* and *Supplementary file 2*). Cluster three includes genes that are enriched in naïve hTSCs, such as *CCR7*, *CTNNB1*, *ELF5*, *EPCAM*, *TFAP2C*, *ITGA6*, *OVOL1*, and *TP63* (*Figure 4D* and *Supplementary file 2*), and is enriched in GO terms associated with cytoskeletal remodeling (*Supplementary file 3*). Cluster four includes genes such as *IGF1R*, *IGF2R*, and *KRT7* that are enriched in both naïve hTSCs and EVTs (*Figure 4D* and *Supplementary file 2*). Finally, clusters 5 and 6 include genes that are enriched in either EVTs or STBs derived from naïve hTSCs, respectively. EVT-specific genes include *FN1*, *HLA-G*, *ITGA1*, *ITGA5*, and *MMP2*, while STB-specific genes include *CGA*, *CGB1*, *INHA*, *PSG1*, *SDC1*, and *TBX3* (*Figure 4D* and *Supplementary file 2*). Notably, *CGA* and *CGB1* encode the α and β chains of the placental hormone hCG, while TBX3 was recently identified as a transcriptional regulator of STB differentiation (*Lv et al., 2019*). EVT-specific genes are enriched in GO terms related to cell migration and invasion, while STB-specific genes are enriched in terms related to membrane fusion and interferon signaling (*Supplementary file 3*). Many, if not all, of the genes highlighted in these clusters have the same specific gene expression profiles reported for primary hTSCs, EVTs, and STBs derived from human blastocysts (*Okae et al., 2018*). We conclude that the derivation of hTSCs from naïve hPSCs and their subsequent, cell-type-specific differentiation recapitulate the processes of hTSC derivation from the human embryo and its differentiation into EVT and STB cell types.

We examined whether hTSCs generated by direct derivation from naïve hPSCs share molecular signatures with human trophectoderm (TE) at pre- or post-implantation stages. In a recent study, human embryos were cultured in vitro past implantation, and single cell RNA-seq (scRNA-seq) was performed at days 6, 8, 10, and 12 post-fertilization (*Zhou et al., 2019*). This work identified genes that are specifically expressed in the human TE, EPI, and primitive endoderm (PE) lineages (*Zhou et al., 2019*). We assayed the expression of these genes in naïve and primary hTSCs, then correlated their expression patterns to that of human TE, EPI, and PE at distinct time points (*Figure 4E*; *Figure 4—figure supplement 1B*). We found that hTSCs cluster the most closely amongst themselves, as well as with day 10 and day 12 TE (*Figure 4E*), while showing the best correlation with day 12 TE (*Figure 4—figure supplement 1B*). In contrast, the hTSCs correlate poorly with the EPI or PE lineages (*Figure 4E*). To further uncover the genes and biological processes that contribute to the similarities between naïve hTSCs and post-implantation TE, we selected those genes that are specifically expressed in the TE or EPI at day 10 or day 12, and examined their expression in naïve hTSCs relative to naïve hPSCs (*Zhou et al., 2019*). We observed significant correlation between naïve hTSC/naïve hPSC in vitro, and TE/EPI at day 10 or day 12 in vivo (r = 0.61) (*Figure 4—figure supplement 1C*). Additionally, we performed GO term analysis on the top 10% most highly upregulated genes in naïve hTSCs relative to naïve hPSCs (*Figure 4—figure supplement 1C* and *Supplementary file 4*). These genes include placenta-associated factors such as *CGB7*, *EGFR*, *ITGA2*, *LHB* and multiple pregnancy-specific glycoproteins (*PSG2-6*), and are enriched in biological processes such as female pregnancy, reproductive processes, and positive regulation of NF-kappa B transcription factor activity (*Figure 4—figure supplement 1D*). This provides additional validation that the process of hTSC derivation from naïve hPSCs activates genes and pathways that are relevant

to human trophoblast biology. We conclude that hTSCs derived from naïve hPSCs are comparable to *bona fide*, blastocyst-derived hTSCs, and are transcriptionally the most similar to human post-implantation TE at day 12.

Finally, we profiled the chromatin accessibility landscape of two naïve-hPSC derived hTSC lines and a blastocyst-derived hTSC line, BT5 (*Okae et al., 2018*), using the assay for transposase-accessible chromatin followed by sequencing (ATAC-seq). Overall, we found ATAC-seq peaks in all three hTSC lines to be highly correlated globally (*Figure 4F*), which indicates that derivation of hTSCs from naïve hPSCs induces a similar epigenomic profile as hTSC derivation from the human blastocyst. We identified differentially accessible regions (DARs) between naïve hESCs (9,059) and naïve hTSCs (12,132), and found that naïve hESC-specific open chromatin sites are more enriched over promoter regions (1,084) and tend to be in closer proximity to the nearest transcriptional start site relative to hTSC-specific open chromatin sites (*Figure 4—figure supplement 2A,B*). In hTSCs, only 208 DARs are located in gene promoter regions, and a higher percentage of hTSC-specific DARs are located in intronic and intergenic regions (*Figure 4—figure supplement 2A,B*), suggesting the importance of distal regulatory elements during differentiation of naïve hPSCs into hTSCs. We further analyzed the ATAC-seq signal at the promoter regions of genes within differentially expresses genes (DEG) cluster 1 and 3 (*Figure 4C*), which are genes more specifically expressed in naïve hPSCs and hTSCs, respectively. Consistent with DAR analysis, we noticed that genes in clusters 1 and 3 are associated with higher ATAC-seq signals at the promoter regions in naïve hESCs and hTSCs, respectively. However, the ATAC-seq signal at the promoter regions of cluster 3 (hTSC-specific) genes was only slightly increased in naïve hTSCs relative to naïve hESCs (*Figure 4—figure supplement 2C*). This indicates that the promoter regions of hTSC-specific genes are already accessible in naïve hESCs, which is consistent with previous findings that naïve hPSCs share open chromatin regions with first-trimester placental tissues (*Pontis et al., 2019*).

To identify transcription factors that may differentially regulate naïve hESCs vs. naïve hTSCs, we examined enrichment for known transcription factor motifs in differentially accessible ATAC peaks. Naïve hESC-specific enrichment was observed for the POU5F1:SOX2 and KLF4 motifs (*Figure 4G*), consistent with the roles of these transcription factors in regulating naïve pluripotency (*Guo et al., 2009*; *Stuart et al., 2019*; *Wong et al., 2016*). Intriguingly, we observed significant enrichment for motifs associated with TEAD4, CEBP, GATA, and TFAP2 over non-promoter regions in naïve hTSCs, while motifs associated with TFAP2, GATA, SP1, ELK1, and GRHL were enriched over promoter regions in naïve hTSCs (*Figure 4G*). Similar motif enrichments were observed for naïve hESCs vs. BT5 hTSC DARs (*Figure 4G*). Among these transcription factors, TEAD4 and TFAP2 are known to be expressed in human TE (*Petropoulos et al., 2016*), while CEBP and GATA factors have been implicated as regulators of trophoblast-specific HLA-G expression at the maternal-fetal interface (*Ferreira et al., 2016*).

The strong enrichment of the TEAD4 binding motif in hTSCs is especially significant, as it suggests that TEAD4 may play a significant role during TE specification in both mouse and human (*Nishioka et al., 2009*; *Nishioka et al., 2008*; *Yagi et al., 2007*). To explore the role of TEAD4 in hTSC derivation from naïve hPSCs, we treated the cells with Verteporfin, an inhibitor that disrupts YAP-TEAD4 interactions and was recently shown to prevent cavity formation in blastocyst-like structures generated from mouse extended pluripotent stem (EPS) cells (*Li et al., 2019*). Verteporfin treatment induced widespread cell death within several days when applied during hTSC derivation (*Figure 4—figure supplement 2D*). We then tested the effect of Verteporfin on the maintenance of naïve hPSCs and found that it had a similar detrimental effect on viability. This suggests that YAP-TEAD4 signaling may be important for both the maintenance of naïve hPSCs and their transition into a trophoblast fate. On the other hand, these experiments may also indicate that Verteporfin is simply toxic at the examined concentration. A recent study has pointed to species-specific differences in requirements for YAP-TEAD4 signaling between mouse and human. Unlike in the mouse embryo, where YAP is nuclear in the TE and cytoplasmic in the EPI compartment (*Nishioka et al., 2009*), YAP is nuclear in both the TE and EPI in human blastocysts (*Qin et al., 2016*). We confirmed by immunofluorescence analysis that YAP shows nuclear localization in both naïve hPSCs and hTSCs (*Figure 4—figure supplement 2E*). Thus, YAP-TEAD4 may contribute to both EPI and TE development in human, potentially by targeting different regulatory elements and transcription factors. In support of this interpretation, we detected ATAC-seq peaks containing TEAD4 transcription factor binding motifs at key pluripotency regulators in naive hESCs, including *KLF4*, *NANOG*, and *DPPA2/4*

(*Figure 4C*; *Supplementary file 5*). Meanwhile, naïve hTSC-specific DARs containing TEAD4 binding motifs are enriched at loci encoding key trophoblast marker genes such as *ITGA2*, *KRT7*, and *EGFR* (*Supplementary file 6*), suggesting that TEAD4 is involved in hTSC specification. Hence, the transcription factor binding motifs that become enriched at open chromatin sites during hTSC derivation from naïve hESCs provide a resource for identifying candidate regulators of early trophoblast specification.

## Discussion

In this study, we have demonstrated that naïve, but not primed, hPSCs can give rise to self-renewing hTSCs capable of undergoing further differentiation into specialized trophoblast cell types. Furthermore, by performing RNA-seq, we have shown that naïve hPSC-derived hTSCs - 'naïve hTSCs' are comparable with blastocyst-derived hTSCs and share the strongest transcriptional correlation with human TE at day 12 post-fertilization. By performing ATAC-seq analyses on hTSCs for the first time, we established that naïve hTSCs and human blastocyst-derived hTSCs have a similar chromatin accessibility landscape, and identified transcription factor binding motifs at sites that become accessible during hTSC derivation. We believe that this work has three major implications, as outlined below.

First, our results give insight into the lineage potential of naïve hPSCs. We previously reported that naïve hPSCs exhibit expression of transcription factors and open chromatin sites associated with the trophoblast lineage (*Pontis et al., 2019*; *Theunissen et al., 2016*). Our data now provide functional evidence that naïve hPSCs indeed have enhanced potential to access trophoblast fates during both spontaneous differentiation assays and upon treatment with recently devised conditions for hTSC isolation (*Okae et al., 2018*). While naïve hPSCs can directly give rise to bipotent hTSCs, primed hPSCs instead give rise to cells with a neuroectodermal gene expression profile when exposed to the same signaling milieu. This is reminiscent of the mouse paradigm where CDX2 transgene expression allowed for mouse trophoblast stem cell (mTSC) differentiation from naïve mESCs, but not from primed mouse epiblast stem cells (mEpiSCs) (*Blij et al., 2015*). We propose that the enhanced, transgene-independent extraembryonic potential of naïve hPSCs is a distinctive functional attribute of naïve human pluripotency, which may reflect the co-expression of embryonic and extraembryonic genes in the human pre-implantation embryo (*Petropoulos et al., 2016*). This conclusion is further corroborated by recent work demonstrating that naïve, but not primed, hPSCs can give rise to human extraembryonic endoderm (XEN) cells in response to XEN media (*Linneberg-Agerholm et al., 2019*). It will be instructive to define the exact time window when extraembryonic differentiation potential is acquired or lost during the interconversion between naïve and primed states. The capacity to attain an hTSC-like state does not, however, appear to be restricted to the naïve state. hTSC-like cells have also been obtained from human expanded potential stem cells (hEPSCs) (*Gao et al., 2019*), while another study reported that putative hTSCs can be derived from primed hPSCs after transition into a CTB-like state (*Mischler et al., 2019*). It will be important to compare the quality of hTSCs derived from these distinct sources, and define transcriptional and/or epigenomic similarities between the various cell types that are capable of differentiation into hTSCs. Such a comparative analysis may reveal the precise molecular characteristics that impart competence for hTSC derivation.

Second, our study provides a cellular model system of early mechanisms governing human trophoblast specification. Previous methodologies for directing human trophoblast differentiation from pluripotent cells have relied on primed hPSCs treated with BMP4 (*Amita et al., 2013*; *Horii et al., 2016*; *Xu et al., 2002*). However, primed hPSCs are most closely aligned with the late post-implantation epiblast based on scRNA-seq studies (*Nakamura et al., 2016*), when TE and EPI lineages have already segregated. Primed hPSCs also have substantially elevated levels of DNA methylation compared to pre-implantation embryos, naïve hPSCs, or hTSCs (*Okae et al., 2018*; *Smith et al., 2014*; *Theunissen et al., 2016*; *Zhou et al., 2019*). Hence, it has been proposed that putative CTB progenitors derived from primed hPSCs upon BMP4 treatment may actually correspond to an extraembryonic mesoderm identity (*Bernardo et al., 2011*). Recent work from the Parast lab confirms that BMP4 treatment alone induces a mixture of trophoblast and mesoderm fates in primed hPSCs (*Horii et al., 2019*). Our data indicate that naïve hPSCs are unresponsive to BMP4 (*Figure 1C*), in accordance with recent findings that naïve hPSCs are recalcitrant to direct application of some

lineage-inductive cues (*Rostovskaya et al., 2019*), and may in fact suggest that the BMP4-mediated differentiation protocol is primed-state specific. However, unlike primed hPSCs, naïve hPSCs when cultured in hTSC media can directly give rise to hTSCs that share transcriptional signatures with human post-implantation TE (*Figures 2* and *4*). We propose that the derivation of hTSCs from naïve hPSCs presents a novel experimental paradigm to identify early determinants of human trophoblast specification.

Our analysis of transcription factor binding motifs in open chromatin regions enriched during hTSC derivation provides a blueprint for exploring candidate regulators of this transition. Intriguingly, the TEAD4 binding motif is significantly enriched at the open non-promoter regions specific to naïve and primary hTSCs. This suggests that TEAD4 plays an important role in human TE development. It may also indicate that the HIPPO/YAP signaling pathway, a known regulator of TEAD4 (*Zhao et al., 2008*), is involved in mouse as well as human TE specification (*Nishioka et al., 2009*; *Nishioka et al., 2008*; *Yagi et al., 2007*). However, unlike in the mouse blastocyst, YAP is also localized to the nucleus in human EPI cells (*Qin et al., 2016*), and we show here that YAP is nuclear in both naïve hPSCs and hTSCs. This points to a possible species-specific difference with HIPPO/YAP/TEAD4 signaling being involved in both TE and EPI development in human. This divergence is not unprecedented: TFAP2C, a trophoblast-specific transcription factor in mouse, was shown to play an essential role in naïve human pluripotency as well (*Pastor et al., 2018*). How HIPPO/YAP/TEAD4 signaling differentially regulates EPI versus TE development in human, and how that differs from the mouse paradigm, will be of interest for future studies.

Third, our work provides a robust methodology for a renewable, patient-specific source of hTSCs. Thus far, the isolation of hTSCs has required the use of blastocysts or first-trimester placental tissues (*Okae et al., 2018*), neither of which are readily accessible or amenable to genetic manipulation. In contrast, naïve hPSCs can be conveniently derived from any pre-existing line of hPSCs and can undergo efficient genetic modification (*Yang et al., 2016*). Thus, naïve hPSCs provide a scalable source of cellular material for hTSC derivation. Furthermore, several groups have recently described protocols for direct reprogramming of human somatic cells into naïve pluripotency (*Giulitti et al., 2019*; *Kilens et al., 2018*; *Liu et al., 2017*; *Wang et al., 2018*). The derivation of naïve iPSCs from patient-specific cells and their subsequent differentiation into hTSCs may provide a pathway to uncover the genetic origins of common pathologies afflicting the trophoblast lineage, such as miscarriage, pre-eclampsia, and fetal growth restriction.

# Materials and methods

## Key resources table

| Reagent type (species) or resource | Designation | Source or reference | Identifiers | Additional information |
|---|---|---|---|---|
| Antibody | anti-KRT7 (Rabbit monoclonal) | Cell signaling | 4465 | 1:100 |
| Antibody | anti-TP63 (Goat monoclonal) | R and D system | AF1916 | 1:20 |
| Antibody | anti-TEAD4 (Mouse monoclonal) | Abcam | ab58310 | 1:400 |
| Antibody | anti-SDC1 (Mouse monoclonal) | Abcam | ab34164 | 1:100 |
| Antibody | anti-ZO1 (Mouse monoclonal) | Invitrogen | 33–9100 | 1:100 |
| Antibody | anti-hCG (Mouse monoclonal) | Invitrogen | 14650882 | 1:100 |
| Antibody | anti-MMP2 (Rabbit monoclonal) | Cell Signaling | 40994 | 1:800 |
| Antibody | anti-SUSD2-PE | BioLegend | 327406 | 1:100 |
| Antibody | anti-CD75-eFluor 660 | Thermo Fisher | 50-0759-42 | 1:100 |
| Antibody | anti-ITG6-FITC | Miltenyi | 130-097-245 | 1:100 |

*Continued on next page*

*Continued*

| Reagent type (species) or resource | Designation | Source or reference | Identifiers | Additional information |
|---|---|---|---|---|
| Antibody | anti-EGFR-APC | BioLegend | 352905 | 1:20 |
| Antibody | anti-HLA-G-PE | Abcam | ab24384 | 1:100 |
| Antibody | anti-YAP (Rabbit monoclonal) | Cell Signaling | 14074 | 1:100 |
| Antibody | anti-mouse-Alexa 488 | Invitrogen | A-21202 | 1:500 |
| Antibody | anti-rabbit-Alexa 647 | Invitrogen | A-31573 | 1:500 |
| Antibody | anti-goat-Alexa 555 | Invitrogen | A-21432 | 1:500 |
| Cell line (Human) | H9 hESC | WashU GEiC | | |
| Cell line (Human) | AN_1.1 iPSC | WashU GEiC | | |
| Cell line (Human) | WIBR3 hESC | Whitehead Institute | | Dr. Jaenisch |
| Cell line (Human) | BT5 hTSC | (*Okae et al., 2018*) | | Drs. Okae, Arima, and Pastor |
| Recombinant protein | EGF | Rockland | 009–001 C26 | |
| Recombinant protein | NRG1 | Cell signaling | 5218SC | |
| Recombinant protein | Activin A | Peprotech | 120–14 | |
| Recombinant protein | BMP4 | R and D Systems | 314 BP | |
| Chemical compound | CHIR99021 | Stemgent | 04–0004 | |
| Chemical compound | A83-01 | BioVision | 1725 | |
| Chemical compound | SB431542 | BioVision | 1674 | |
| Chemical compound | VPA | Tocris | 2815 | |
| Chemical compound | Forskolin | Sigma-Aldrich | F3917 | |
| Chemical compound | Y-27632 | Stemgent | 04–0012 | |
| Chemical compound | PD0325901 | Stemgent | 04–0006 | |
| Chemical compound | IM-12 | Enzo | BML-WN102 | |
| Chemical compound | SB590885 | Tocris | 2650 | |
| Chemical compound | WH4-023 | A Chemtek | H620061 | |
| Chemical compound | XAV939 | Selleckchem | S1180 | |
| Chemical compound | Gö6983 | Tocris | 2285 | |
| Chemical compound | Verteporfin | Tocris | 5305 | |
| Commercial assay, kit | MMP2 ELISA kit | Abcam | ab100606 | |

## Cell line verification and testing

The identities of the cell lines used in this study were authenticated using Short Tandem Repeat (STR) profiling. The cell culture is regularly tested and negative for mycoplasma contamination.

## Culture of naïve and primed hPSCs

Primed hPSCs were cultured in mTeSR1 (STEMCELL Technologies, 85850) on Matrigel (Corning, 354277) coated wells and passaged using ReLeSR (STEMCELL Technologies, 05872) every 4 to 6 days. Primed hPSCs were cultured in 5% $CO_2$ and 20% $O_2$. Naive hPSCs were cultured on mitomycin C-inactivated mouse embryonic fibroblast (MEF) feeder cells, and were passaged by a brief PBS wash followed by single-cell dissociation using 5 min treatment with TrypLE Express (Gibco, 12604) and centrifugation in fibroblast medium [DMEM (Millipore Sigma, #SLM-021-B) supplemented with 10% FBS (Millipore Sigma, ES-009-B), 1X GlutaMAX (Gibco, 35050), and 1% penicillin-streptomycin (Gibco, 15140)]. Naive hPSCs were cultured in the 5i/L/A media as previously described (*Theunissen et al., 2014*). 500 mL of 5i/L/A was generated by combining: 240 mL DMEM/F12 (Gibco, 11320), 240 mL Neurobasal (Gibco, 21103), 5 mL N2 100X supplement (Gibco, 17502), 10

mL B27 50X supplement (Gibco, 17504), 10 µg recombinant human LIF (made in-house), 1X Gluta-MAX, 1X MEM NEAA (Gibco, 11140), 0.1 mM β-mercaptoethanol (Millipore Sigma, 8.05740), 1% penicillin-streptomycin, 50 µg/ml BSA Fraction V (Gibco, 15260), and the following small molecules and cytokines: 1 µM PD0325901 (Stemgent, 04–0006), 1 µM IM-12 (Enzo, BML-WN102), 0.5 µM SB590885 (Tocris, 2650), 1 µM WH4-023 (A Chemtek, H620061), 10 µM Y-27632 (Stemgent, 04–0012), and 10 ng/mL Activin A (Peprotech, 120–14). Naïve hPSCs were cultured in 5% $O_2$, 5% $CO_2$. For indicated experiments, 1 or 2 µM Verteporfin was added to 5i/L/A. For primed to naïve hPSC conversion, $2 \times 10^5$ single primed cells were seeded on a 6-well plate with MEF feeder layer in 2 mL mTeSR1 supplemented with 10 µM Y-27632. Two days later, medium was switched to 5i/L/A. After 7 to 10 days from seeding, the cells were expanded polyclonally using TrypLE Express on a MEF feeder layer. Tissue culture media were filtered using a 0.22 µm filter. Media were changed every 1–2 days. Naïve hPSCs before passage 10 were used for experiments.

For naïve hPSC clonal expansion experiments, naïve cells were passaged and seeded at clonal density (ca. 1,000 cells per well of a 6-well plate). Single cell clones that exhibit typical dome-shaped morphology were picked and single cell dissociated. They were then seeded and expanded on MEF feeder layer in 5i/L/A as described above.

For indicated experiments, naïve hPSCs were cultured in PXGL [N2B27 supplemented with 1 µM PD0325901, 2 µM XAV939 (Selleckchem, S1180), 2 µM Gö6983 (Tocris, 2285), and 10 ng/mL human LIF] as previously described (*Bredenkamp et al., 2019b*). 10 µM Y-27632 was added during passaging. Primed hPSCs were first converted to the naïve state using 5i/L/A, then the media was switched to PXGL for eight passages before the cells were used for experiments. Cells were cultured on inactivated MEF feeder layers and in 5% $O_2$, 5% $CO_2$. Media were changed every 1–2 days.

## Capacitation of naïve hPSCs

Capacitation of naïve hPSCs was performed as previously described (*Rostovskaya et al., 2019*). Briefly, around $0.5 \times 10^6$ naïve hPSCs were seeded on one well of a Geltrex (Thermo Fisher, A1413201) coated 6-well plate. The cells were first cultured in 5i/L/A for 2 days, then cultured in capacitation media [N2B27 supplemented with 2 µM XAV939] for 10 more days. Media were changed every 1–2 days. The cells were passaged once using TrypLE Express during capacitation at a ratio of 1:2 when confluent, usually after 4 or 5 days of culture in capacitation media. 10 µM of Y-27632 was added for 24 hr following passaging. After 10 days of treatment with capacitation media, the cells were used for subsequent analysis.

## Embryoid body formation

hPSCs were single-cell dissociated using TrypLE, and $3.0 \times 10^6$ cells were aggregated in EB media [DMEM/F12 supplemented with 20% FBS, 1X GlutaMax, 1X MEM NEAA, 0.1 mM β-mercaptoethanol, and 1% penicillin-streptomycin] supplemented with 10 µM Y-27632 in 24-well 800 µm Aggrewell plates (STEMCELL Technologies, 34811). After 24 hr, the aggregates were flushed from the Aggrewell using EB media, and cultured on ultra-low attachment 6-well plates in EB media only. Media were changed every 2 days. After 12 days of culture, the EBs were collected for downstream analysis.

## BMP4-directed trophoblast differentiation

Trophoblast differentiation from hPSCs using BMP4 was performed as previously described (*Horii et al., 2016*). Prior to differentiation, primed or naïve hPSCs were adapted to culture in StemPro (Thermo Fisher, A1000701) medium with 12 ng/mL recombinant basic FGF (bFGF) (Thermo Fisher, PHG0261) on Geltrex coated plates. Naïve and capacitated hPSCs were directly subjected to trophoblast differentiation conditions. Briefly, $2.0 \times 10^4$ to $1.0 \times 10^5$ primed, re-primed, naïve, or capacitated hPSCs were seeded on Geltrex coated 24-well plates in EMIM medium [DMEM/F12 supplemented with 1X MEM NEAA, 2% BSA (Sigma-Aldrich, A9418), 2 mM L-Glutamine (Corning, 25–005 CI), 1% ITS (Gibco, 41400), and 100 ng/mL heparin sulfate (Stemcell Technologies, 07980] for 2 days. They were cultured in EMIM supplemented with 10 ng/mL human BMP4 (R and D Systems, 314 BP) for four more days, at which point they were designated as CTBs. For terminal EVT and STB differentiation, the CTBs were further cultured in FCM [DMEM/F12 supplemented with 1X Gluta-MAX, and 1% non-essential amino acid, 0.1 mM β-mercaptoethanol; conditioned on irradiated MEFs

for 24 hr] supplemented with 10 ng/mL human BMP4 for 6 days. Media were changed every 1–2 days. Cells were cultured in 5% $CO_2$ and 20% $O_2$.

## Culture of hTSCs

hTSCs were cultured as previously described (Okae et al., 2018). Briefly, a 6-well plate was coated with 5 μg/mL Collagen IV (Corning, 354233) at 37°C overnight. Cells were cultured in 2 mL TS medium [DMEM/F12 supplemented with 0.1 mM 2-mercaptoethanol, 0.2% FBS, 0.5% Penicillin-Streptomycin, 0.3% BSA, 1% ITS-X (Gibco, 51500), 1.5 μg/ml L-ascorbic acid (Wako, 013–12061), 50 ng/ml EGF (Rockland, 009–001 C26), 2 μM CHIR99021 (Stemgent, 04–0004), 0.5 μM A83-01 (BioVision, 1725), 1 μM SB431542 (BioVision, 1674), 0.8 mM VPA (Tocris, 2815), and 5 μM Y-27632] and in 5% $CO_2$ and 20% $O_2$. Media were changed every 2 days, and cells were passaged using TrypLE Express every 3 days at a ratio of 1:4. Unless otherwise specified, hTSCs between passage 10 and 20 were used for experiments.

## Derivation of hTSCs from naïve hPSCs

Naïve and primed hPSCs were single-cell dissociated by TrypLE Express, and $0.5–1.0 \times 10^6$ cells were seeded in a 6-well plated pre-coated with 5 μg/mL Collagen IV and cultured in 2 mL TS medium. Cells were cultured in 5% $CO_2$ and 20% $O_2$, media was changed every 2 days, and passaged upon 80–100% confluency at a ratio of 1:2 to 1:4. For indicated experiments, 1 or 2 μM Verteporfin was added to TS medium during the derivation process. For derivation from naïve hPSCs, the cells grew slowly during the initial few passages. Between passage 5 and 10, highly proliferative hTSCs emerged. In contrast, for derivation from primed hPSCs, the cells did not appear to gain any hTSC characteristic even following 20+ passages.

## EVT and STB differentiation from hTSCs

Differentiation of hTSCs into terminal cell types were performed as previously described (Okae et al., 2018), with minor modifications. Prior to differentiation, hTSCs were grown to about 80% confluency, and then single-cell dissociated using TrypLE Express. For EVT differentiation, 6-well plates were coated with 1 μg/mL Collagen IV overnight. $0.75 \times 10^5$ hTSCs were seeded per well in 2 mL EVT basal medium [DMEM/F12 supplemented with 0.1 mM β-mercaptoethanol, 0.5% penicillin-streptomycin, 0.3% BSA, 1% ITS-X, 7.5 μM A83-01, 2.5 μM Y27632] supplemented with 4% KSR (Gibco, 10828) and 100 ng/mL NRG1 (Cell signaling, 5218SC). Matrigel was added to a 2% final concentration shortly after resuspending hTSCs in the medium. At day 3, the media were replaced with 2 mL EVT basal medium supplemented with 4% KSR, and Matrigel was added to a 0.5% final concentration. At day 6, the media were replaced with 2 mL EVT basal medium, and Matrigel was added to a 0.5% final concentration. At day 8, the cells were ready for downstream analysis.

For 2D STB differentiation, 6-well plates were coated with 2.5 μg/mL Collagen IV overnight. $1 \times 10^5$ hTSCs were seeded per well in 2 mL 2D STB medium [DMEM/F12 supplemented with 0.1 mM β-mercaptoethanol, 0.5% penicillin-streptomycin, 0.3% BSA, 1% ITS-X, 2.5 μM Y-27632, 2 μM Forskolin (Sigma-Aldrich, F3917), and 4% KSR]. The media was changed at day 3, and at day 6 the cells were ready for downstream analysis.

For 3D STB differentiation, $2.5 \times 10^5$ hTSCs were seeded per well in 3 mL 3D STB medium [DMEM/F12 supplemented with 0.1 mM β-mercaptoethanol, 0.5% penicillin-streptomycin, 0.3% BSA, 1% ITS-X, 2.5 μM Y-27632, 50 ng/ml EGF, 2 μM Forskolin, and 4% KSR] in an ultra-low attachment 6-well plate. At day 3, another 3 mL of 3D STB medium was added per well. At day 6, the cells were passed through a 40 μm cell strainer, and the cells remaining on the strainer were collected and used for downstream analysis.

## Immunofluorescence staining

Cells were fixed with 4% paraformaldehyde for 20 min at room temperature, then washed with PBS three times. The 3D STBs were resuspended in a small amount of PBS and seeded on plus-charged slides. PBS was allowed to air dry, and immunostaining was performed directly on the slides. For other types of cells, immunostaining was performed directly in the wells. The cells were permeabilized with 0.1% Triton X-100 (Sigma, T8787) in PBS for 5 min, then blocked with blocking buffer [PBS supplemented with 0.5% BSA and 0.1% Triton X-100] for one hour. Cells were then incubated with

primary antibodies diluted in the blocking buffer overnight at 4°C. The following primary antibodies were used: anti-KRT7, 1:100 (Cell signaling, 4465); anti-TP63, 1:20, (R and D system, AF1916); anti-TEAD4, 1:400 (Abcam, ab58310); anti-SDC1, 1:100 (Abcam, ab34164); anti-ZO1, 1:100 (Invitrogen, 33–9100); anti-hCG, 1:100 (Invitrogen, 14650882); anti-MMP2, 1:800 (Cell Signaling, 40994); anti-YAP, 1:100 (Cell signaling, 14074). The cells were washed 3 times in PBS, then incubated with secondary antibodies diluted in blocking buffer for 1 hr at room temperature. The following secondary antibodies were used: anti-mouse-Alexa 488, 1:500 (Invitrogen, A-21202); anti-rabbit-Alexa 647, 1:500 (Invitrogen, A-31573); anti-goat-Alexa 555, 1:500 (Invitrogen, A-21432). The nuclei were stained with Hoechst 33258 (Thermo Fisher, H3569). Cells were washed 3 times in PBS, then imaged with a Leica DMi-8 fluorescence microscope. Some images were globally adjusted for brightness and/or contrast.

## Flow cytometry

Cells were single-cell dissociated using TrypLE Express and washed once in FACS buffer [PBS supplemented with 5% FBS]. The cells were then resuspended in 100 μL fresh FACS buffer, and incubated with antibodies for 30 min on ice. The following antibodies were used: anti-SUSD2-PE, 1:100 (BioLegend, 327406); anti-CD75-eFluor 660, 1:100 (Thermo Fisher, 50-0759-42); anti-ITG6-FITC, 1:100 (Miltenyi, 130-097-245); anti-EGFR-APC, 1:20 (BioLegend, 352905); anti-HLA-G-PE, 1:100 (Abcam, ab24384). Following antibody incubation, the cells were washed once with FACS buffer, resuspended in fresh FACS buffer, and passed through a cell strainer. Unstained cells that have undergone the same procedures were used as controls. Flow cytometry was performed using a BD LSRFortessa X-20, and the data were analyzed using the FlowJo software.

## Measurement of MMP2 levels in media

At day 8 of EVT differentiation, the media supernatants which had been in culture for 2 days were collected, and stored at −80°C. As controls, $0.75 \times 10^5$ hTSCs were seeded per well in 2 mL TS medium. After 2 days of culture, the media supernatants were collected and stored at −80°C. The amount of secreted MMP2 was measured using a commercial MMP2 ELISA kit (Abcam, ab100606). Each experiment was performed with two biological replicates, and each biological replicate contains two technical replicates.

## Quantitative real-time PCR

Total RNA was isolated using the E.Z.N.A. total RNA kit I (Omega, D6834), and cDNA synthesis was performed from total RNA using the high capacity cDNA reverse transcription kit (Applied Biosystems, 4368814). Real-time PCR was performed using PowerUp SYBR green master mix (Applied Biosystems, A25743) on the StepOnePlus Real-Time PCR System (Applied Biosystems). All analyses were done in triplicate. Gene expression was normalized to RPLP0. Error bars represent the standard deviation (SD) of the mean of triplicate reactions. Student's t test was performed for statistical analysis. Primer sequences are included in the following primer table:

| Gene | Primer sequence (5'- 3') |
| --- | --- |
| RPLP0-F | GCTTCCTGGAGGGTGTCC |
| RPLP0-R | GGACTCGTTTGTACCCGTTG |
| KLF17-F | CTGCCTGAGCGTGGTATGAG |
| KLF17-R | TCATCCGGGAAGGAGTGAGA |
| Pax6-F | CTTTGCTTGGGAAATCCGAG |
| Pax6-R | AGCCAGGTTGCGAAGAACTC |
| VIM-F | TGTCCAAATCGATGTGGATGTTTC |
| VIM-R | TTGTACCATTCTTCTGCCTCCTG |
| Sox17-F | CGCACGGAATTTGAACAGTA |
| Sox17-R | GGATCAGGGACCTGTCACAC |

*Continued on next page*

*Continued*

| Gene | Primer sequence (5'- 3') |
|------|--------------------------|
| CDX2-F | TTCACTACAGTCGCTACATCACC |
| CDX2-R | TTGATTTTCCTCTCCTTTGCTC |
| TEAD4-F | CAGGTGGTGGAGAAAGTTGAGA |
| TEAD4-R | GTGCTTGAGCTTGTGGATGAAG |
| TFAP2C-F | TCTTGGAGGACGAAATGAGATGG |
| TFAP2C-R | GGGCTTCTTTGATGTAGTTCTGC |
| ELF5-F | AGTCTGCACTGACATTTTCTCATC |
| ELF5-R | CAGAAGTCCTAGGGGCAGTC |
| KRT7-F | AGGATGTGGATGCTGCCTAC |
| KRT7-R | CACCACAGATGTGTCGGAGA |
| GATA3-F | TGCAGGAGCAGTATCATGAAGCCT |
| GATA3-R | GCATCAAACAACTGTGGCCAGTGA |
| MMP2-F | TGGCACCCATTTACACCTACAC |
| MMP2-R | ATGTCAGGAGAGGCCCCATAGA |
| HLA-G-F | CAGATACCTGGAGAACGGGA |
| HLA-G-R | CAGTATGATCTCCGCAGGGT |
| CGB-F | ACCCTGGCTGTGGAGAAGG |
| CGB-R | ATGGACTCGAAGCGCACA |
| SDC1-F | GCTGACCTTCACACTCCCCA |
| SDC1-R | CAAAGGTGAAGTCCTGCTCCC |

## Matrigel invasion assay

In vitro invasion assay was performed in Matrigel-coated transwell inserts with 8.0 µm pores (Corning, 354480). EVTs or hTSCs were single cell dissociated, and seeded at density of $2 \times 10^5$ cells per well into the upper chamber of Matrigel-coated transwells in 200 µl EVT basal medium or TS medium. The lower chamber was filled with 800 µl of the same type of medium containing 20% FBS. Cells were cultured at 37°C in 5% $CO_2$ and 20% $O_2$. After 36 hr, cells on the upper chamber were carefully removed with a cotton swab. The lower chamber was fixed with 4% paraformaldehyde, washed with PBS, and then stained with crystal violet. Invaded cells were imaged on a Leica DMi1 microscope. Thereafter, the stained cells from five random fields were counted to calculate the relative fold change in the number of invading cells. Student's t test was performed for statistical analysis. Each experiment was performed in triplicate.

## RNA-seq

Total RNA was isolated using the E.Z.N.A. total RNA kit I. Library construction was performed using the SMARTer Directional cDNA Library Construction Kit (Clontech, 634933). Libraries were sequenced on an Illumina HiSeq3000 $1 \times 50$ platform. RNA-seq reads were aligned to the human genome hg38 with STAR version 2.5.4b (*Dobin et al., 2013*). Gene counts were derived from the number of uniquely aligned unambiguous reads by Subread:featureCount (*Liao et al., 2013*), version 1.4.6, with hg38 gene annotation ENCODE V27 (*Harrow et al., 2012*). All gene-level transcript counts were then imported into the R/Bioconductor package DESeq2 (*Love et al., 2014*). Transcripts with CPM > 1.0 were converted into a DESeq2 dataset and then regularized log transformed using the rlog function from the DESeq2 package. Adjusted p-values for DGE were determined by DESeq2 using the R stats function p.adjust using the Benjamini and Hochberg correction to determine the false discovery rate with a 2- fold expression change and FDR < 0.05 required to consider a gene differentially expressed. Principal Component Analysis was performed using plotPCA also from the DESeq2 package and plotted using ggplot2. The differentially expressed gene lists from pairwise comparisons between conditions were merged along with their RPKMs. RPKMs were

z-score scaled for each gene, and k-means clustering was performed using the kmeans function from the R stats package to separate genes into six clusters with options 'iter.max=1000' and 'nstart = 1000' specified to maximize reproducibility of clustering.

Human pre- and post-implantation embryo single cell gene expression data (transcript per million, TPM) were downloaded from GEO with accession number GSE109555 (*Zhou et al., 2019*). Gene expressions of single cells in the TE, EPI, and PE lineages were further averaged for days 6, 8, 10, and 12 based on cell identification. The bulk RNA-seq gene expressions data of BT5 hTSC, naïve hTSC, and naïve hPSC were scaled to TPM and combined with the human embryo single cell gene expression data based on the common gene symbols. The expression of TE, EPI, and PE marker genes (from Zhou et al. supplemental table 5) in all the samples was isolated to calculate the Pearson correlation between all sample pairs. The correlation coefficients were plotted in R using gplots heatmap.2 function. The gene expression ratios of TE and EPI marker genes in naïve hTSC/naïve hPSC and TE/EPI (days 10 and 12) were calculated and plotted in R using gplots heatmap.2.

## ATAC-seq

ATAC-seq was performed as previously described with minor modifications (*Corces et al., 2017*). Cells were harvested by TrypLE Express dissociation and centrifuged at 500 RCF for 5 min at 4℃. The supernatant was aspirated. Cells were washed once with cold PBS. Cell pellets were then lysed in 100 μL ATAC-seq RSB [10 mM Tris pH 7.4, 10 mM NaCl, 3 mM MgCl$_2$] containing 0.1% NP40, 0.1% Tween-20, and 0.01% Digitonin by pipetting up and down and incubating on ice for 3 min. 1 mL of ATAC-seq RSB containing 0.1% Tween-20 was added and mixed with the lysis reaction. Nuclei were then pelleted by centrifuging at 800 RCF for 5 min at 4℃. Supernatant was removed, and the nuclear pellets were resuspended in 20 μL 2x TD buffer [20 mM Tris pH 7.6, 10 mM MgCl$_2$, 20% Dimethyl Formamide]. Nuclei were counted, and 50,000 counted nuclei were then transferred to a tube with 2x TD buffer filled up to 25 μL. 25 μL of transposition mix [2.5 μL Transposase (100 nM final) (Illumina, 20034197, 16.5 μL PBS, 0.5 μL 1% Digitonin, 0.5 μL 10% Tween-20, 5 μL H2O) was then added. Transposition reactions were mixed and incubated at 37℃ for 30 min with gentle tapping every 10 min. Reactions were cleaned up with the Zymo DNA Clean and Concentrator-5 kit (Zymo Research, D4014). The ATAC-seq library was prepared by amplifying for nine cycles on a PCR machine. The PCR reaction was purified with Ampure XP beads (Beckman Coulter, A63880) using double size selection following the manufacturer's protocol, in which 27.5 μL beads (0.55X sample volume) and 50 μL beads (1.55X sample volume) were used based on 50 μL PCR reaction. The ATAC-seq libraries were quantitated by Qubit assays and sequenced on an Illumina NextSeq platform. QC and analysis on ATAC-seq libraries was performed using AIAP (*Liu et al., 2019*). The generated peaks files for each library were merged using bedtools merge, and counts on each peak were quantified for all libraries using bedtools coverage. Differentially accessible region (DAR) analysis was performed as described above for DEG analysis on RNA-seq libraries with the following modifications: peaks with an average read density < 5 CPM across all libraries were excluded and a significance cutoff of |L2FC| > 1 and padj < $1\times10^{-3}$ was required for a region to be considered differentially accessible. DARs were split based on the direction of increased accessibility and annotated using the annotatePeaks.pl function from HOMER (*Heinz et al., 2010*). Annotated DARs were classified as promoter or non-promoter peaks DARs on their distance to the nearest transcription start site (TSS) using a cutoff of 2 kb from a TSS. Motif analysis was performed on these DARs using the findMotifGenome.pl function from HOMER with the option '-size given'. The transcription factor (TF) motifs were selected if the motif can be identified in at least 10% of input DARs and the matching score of TF motif is > 0.9. The motif occupancy heatmap was plotted in R using pheatmap function. Promoter regions for genes from the RNA-seq k-means clustering were determined by their genomic location in the GENCODE gene annotation file (hg38, V27).

## Acknowledgements

We thank the Genome Technology Access Center in the Department of Genetics at Washington University School of Medicine for help with genomic analysis. The Center is partially supported by NCI Cancer Center Support Grant #P30 CA91842 to the Siteman Cancer Center and by ICTS/CTSA Grant# UL1 TR000448 from the National Center for Research Resources (NCRR), a component of the National Institutes of Health (NIH), and NIH Roadmap for Medical Research. This publication is

solely the responsibility of the authors and does not necessarily represent the official view of NCRR or NIH. We are also grateful to Dr. Rudolf Jaenisch (Whitehead Institute for Biomedical Research) for sharing the WIBR3 hESC line, and Dr. William Pastor (McGill University), Dr. Hiroaki Okae (Tohoku University) and Dr. Takahiro Arima (Tohoku University) for sharing the BT5 hTSC line. We thank Dr. Helen McNeill for providing the YAP antibody and Dr. Angela Bowman for proofreading the manuscript.

## Additional information

### Competing interests

Lilianna Solnica-Krezel: Reviewing editor, *eLife*. The other authors declare that no competing interests exist.

### Funding

| Funder | Grant reference number | Author |
| --- | --- | --- |
| Children's Discovery Institute | CDI-LI-2019-819 | Lilianna Solnica-Krezel Thorold W Theunissen |
| McDonnell Center for Cellular and Molecular Neurobiology | 22-3930-26275D | Thorold W Theunissen |
| NIH Office of the Director | NIH Director's New Innovator Award DP2 GM137418 | Thorold W Theunissen |
| Shipley Foundation | Program for Innovation in Stem Cell Science | Thorold W Theunissen |
| Edward Mallinckrodt, Jr. Foundation | | Thorold W Theunissen |

The funders had no role in study design, data collection and interpretation, or the decision to submit the work for publication.

### Author contributions

Chen Dong, Conceptualization, Data curation, Formal analysis, Validation, Investigation, Visualization, Methodology, Writing - original draft, Project administration, Writing - review and editing; Mariana Beltcheva, Investigation, Writing - review and editing, Helped perform the BMP4-directed trophoblast differentiation experiments; Paul Gontarz, Data curation, Formal analysis; Bo Zhang, Data curation, Formal analysis, Writing - review and editing; Pooja Popli, Formal analysis, Investigation, Performed the Matrigel invasion assay; Laura A Fischer, Investigation, Writing - review and editing, Helped perform cell culture experiments; Shafqat A Khan, Investigation, Helped perform flow cytometry experiments; Kyoung-mi Park, Investigation, Helped perform cell culture experiments; Eun-Ja Yoon, Investigation, Writing - review and editing, Helped perform BMP4-directed trophoblast differentiation experiments; Xiaoyun Xing, Investigation, X.X. performed ATAC-seq sample and library preparation; Ramakrishna Kommagani, Ting Wang, Supervision, Writing - review and editing; Lilianna Solnica-Krezel, Supervision, Funding acquisition, Writing - review and editing; Thorold W Theunissen, Conceptualization, Supervision, Funding acquisition, Project administration, Writing - review and editing

### Author ORCIDs

Chen Dong https://orcid.org/0000-0002-5405-5715
Ramakrishna Kommagani http://orcid.org/0000-0003-0403-0971
Thorold W Theunissen https://orcid.org/0000-0001-6943-7858

### Decision letter and Author response

Decision letter https://doi.org/10.7554/eLife.52504.sa1
Author response https://doi.org/10.7554/eLife.52504.sa2

# Additional files

## Supplementary files

- Supplementary file 1. DEGs between naïve hTSCs and primed hPSCs cultured in hTSC media.
- Supplementary file 2. DEG clusters of naïve hPSCs, naïve hTSCs, EVTs, and STBs.
- Supplementary file 3. GO term analysis of DEG clusters of naïve hPSCs, naïve hTSCs, EVTs, and STBs.
- Supplementary file 4. GO term analysis of the most upregulated genes in naïve hTSC relative to naïve hPSC (*Figure 4—figure supplement 1C*).
- Supplementary file 5. DEG cluster one genes that contain naïve hESC ATAC-seq peaks with TEAD4 transcription factor binding motif.
- Supplementary file 6. DEG cluster 1 to 6 genes that contain naïve hTSC-specific ATAC-seq peaks with TEAD4 transcription factor binding motif.
- Transparent reporting form

## Data availability

The accession number for the RNA-seq and ATAC-seq data is GSE138762.

The following dataset was generated:

| Author(s) | Year | Dataset title | Dataset URL | Database and Identifier |
|---|---|---|---|---|
| Theunissen T, Dong C, Gontarz P, Zhang B, Wang T, Xing X | 2019 | Derivation of trophoblast stem cells from naïve human pluripotent stem cells | https://www.ncbi.nlm.nih.gov/geo/query/acc.cgi?acc=GSE138762 | NCBI Gene Expression Omnibus, GSE138762 |

The following previously published datasets were used:

| Author(s) | Year | Dataset title | Dataset URL | Database and Identifier |
|---|---|---|---|---|
| Zhou F, Wang R | 2019 | Reconstituting the transcriptome and DNA methylome landscapes of human implantation | https://www.ncbi.nlm.nih.gov/geo/query/acc.cgi?acc=GSE109555 | NCBI Gene Expression Omnibus, GSE109555 |
| Pastor WA, Liu W | 2018 | TFAP2C regulates transcription in human naive pluripotency by opening enhancers | https://www.ncbi.nlm.nih.gov/geo/query/acc.cgi?acc=GSE101074 | NCBI Gene Expression Omnibus, GSE101074 |

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
