## [Decision Letter]

**Acceptance summary:**

In this manuscript Dong et al. explore whether human trophoblast stem cells (TSCs) can be obtained by direct differentiation of human embryonic stem cells (ESCs), and whether the initial pluripotent status of the cells determines the competence for TSC specification. To this end the authors compare the response of naïve and primed ESCs to a recently published human TSC medium. Whereas naïve ESCs give rise to putative human TSCs, primed ESCs fail to do so. Further, the authors characterize the gene expression patterns and chromatin accessibility of the human TSCs and propose the naïve ESC – human TSC conversion could be used to study trophectoderm (TE) specification.

**Decision letter after peer review:**

Thank you for submitting your article "Derivation of trophoblast stem cells from naïve human pluripotent stem cells" for consideration by *eLife*. Your article has been reviewed by three peer reviewers, and the evaluation has been overseen by Marianne Bronner as the Senior and Reviewing Editor. The following individual involved in review of your submission has agreed to reveal their identity: Magdalena Zernicka-Goetz (Reviewer #1).

The reviewers have discussed the reviews with one another and the Reviewing Editor has drafted this decision to help you prepare a revised submission.

Summary:

In this manuscript Dong et al. explore whether human TSCs can be obtained by direct differentiation of human ESCs, and whether the initial pluripotent status of the cells determines the competence for TSC specification. To this end they compare the response of naïve and primed ESCs to a recently published human TSC medium. They report that whereas naïve ESCs give rise to putative human TSCs primed ESCs fail to do so. Further, they characterize the gene expression patterns and chromatin accessibility of the human TSCs and propose the naïve ESC – human TSC conversion could be used to study trophectoderm (TE) specification.

Essential revisions:

The majority of the suggestions raised by all three reviewers seem fairly straightforward to address with bioinformatic analyses and changes to the text. In addition, there are several wet-lab experiments that are aimed at firming up the conclusions and digging a bit deeper into what new things can the cell model tell us. We enclose the full reviews for details.

An important point is to establish the final identity of the cells. Human TSCs derived from human blastocysts or placentas following the protocol of Okae et al. acquire a post-implantation character. However, here the authors report that naïve human ESCs treated with Okae's media acquire a pre-implantation character. This conclusion should be supported by comparing the gene expression profile of the authors naïve human TSCs to (1) the canonical human TSCs and (2) to this of the pre-implantation human embryos.

Reviewer #1:

The manuscript is very well written, the data is clearly presented, the conclusions are well supported by the experimental data and the findings will be very relevant to the stem cell community. I consider this paper suitable for publication in *eLife*. I only have a few comments:

- A recent manuscript (Mischler et al., 2019) has reported the formation of human TSCs from primed ESCs. The authors should comment on these findings in relation to their findings. Mischler et al. report that the neuroectodermal differentiation observed when primed human ESCs are used as a starting point is abolished if SP1 is added to the medium. This means that given the appropriate conditions primed human ESCs are also competent for TSC specification.

- The most problematic aspect of this manuscript is the final identity of the cells. Human TSCs derived from human blastocysts or human placentas following the protocol of Okae et al. acquire a cytotrophoblast (post-implantation) character. However, here the authors report that naïve human ESCs treated with Okae's media acquire a TE (pre-implantation) character. To support this conclusion they should compare the gene expression profile of their naïve human TSCs and the canonical human TSCs. Moreover, although they compare the naïve human TSCs to pre-implantation human embryos they are missing the comparison between naïve human TSCs and post-implantation human embryos (Zhou et al., 2019). Are the naïve human TSCs more similar to pre-implantation TE than post-implantation TE?

- The authors report that naïve human ESCs are not readily competent to respond to the BMP4 treatment, but they can respond to the human TSC medium without previous capacitation. This is a very interesting difference and the authors could comment on it further in the Discussion.

- At the end of the Introduction the authors state that they have characterized the human TSCs functionally but no functional experiments (e.g. human-mouse chimeras) are presented. The authors should remove this claim or perform the appropriate experiments.

Reviewer #2:

This study demonstrates for the first time that naïve-state human pluripotent stem cells (PSCs) can generate trophoblast stem cells (TSCs) in culture. The work is important because it provides a compelling demonstration that functional differences exist between naïve and primed states of human pluripotency with naïve cells uniquely having the developmental capability to form TSCs. In addition, this relatively straightforward route to producing TSCs will make it much easier to study human trophoblast and other placental cell types. Overall, the study has been well carried out with appropriate methods and the use of multiple cell lines. The data are clear and mostly convincing. Current weaknesses of the manuscript are that the work stops short of using the TSCs for anything particularly new and interesting, and there are individual areas where the results need to be better supported.

1) The manuscript would clearly be strengthened if the cell model (PSC to TSC) was used to uncover new aspects of stem cell or trophoblast biology. One possibility would be to test at what point along the naïve to primed capacitation (or re-priming) process do the cells lose their capability to derive proliferative TSCs. This would tell us useful information about the timing of lineage restriction events, and this should be relatively straightforward to test. Another possibility is to focus on the role of TEAD4 in TSC specification that the authors allude to at the end of the manuscript. Along these lines, experiments aimed at uncovering the role of TEAD4 through the identification of target sites or using available small molecular inhibitors to test the requirement for TEAD4 function in PSC to TSC conversion could be suitable.

2) In the Introduction, the authors mention that naïve PSCs maintained in 5i/L/A and t2i/L/Gö have transcriptional and epigenetic features of the human preimplantation epiblast. There are also some subtle differences between naïve PSCs cultured in the two conditions, particularly related to the exact stage of human development that show the closest correspondence. Because of this, I think it is important that the authors test whether naïve PSCs maintained in t2i/L/Gö can also generate proliferative TSCs.

3) Although unlikely, it is formally possible that naive PSC cultures contain a mixed population with a small number of pre-existing trophoblast progenitor cells. Can the authors provide evidence to support or refute this possibility? Expanding single-cell naïve PSC clones and demonstrating the capability of these clonally-derived lines to form TSCs would be sufficient.

4) It would be useful to see how the naïve TSCs compare to the blastocyst-derived TSCs. The authors should add the published RNA-Seq data sets from Okae et al. into their PCA plot in Figure 4A.

5) Figure 2E: images are poor quality and it is difficult to tell if there is much signal in the cells.

6) Related to Figure 4I, if you call differentially accessible regions between blastocyst-derived TSCs versus naive PSCs and also between naive TSCs versus naive PSCs, do you get the same TF motifs? (My understanding is that current Figure 4I data show merged data sets).

7) I think some of the text could be better balanced to help interpret the results and to communicate how the various cell types might correspond to embryo development. The authors start out by showing primed PSCs undergo differentiation in response to BMP signalling, whereas naïve PSCs do not. Based on a set of markers, and in line with previous studies, the authors conclude that primed PSCs retain some capacity to form trophoblast. This has never made much sense from a developmental point of view and in some ways detracts from the more novel finding here that naïve PSCs really can make trophoblast. Whether the BMP4-treated primed PSCs are trophoblast is a matter of debate with at least one paper suggesting that they are more similar to extraembryonic mesoderm, which share many of the marker genes to trophoblast (PMID: 21816365). The other point is related to the comparisons between TSCs and preimplantation trophectoderm (TE) later in the manuscript. This is a reasonable comparison to make, but it is very unlikely that TE contain a highly proliferative trophoblast progenitor, and so I think it is important to communicate that the TSCs are not expected to align too closely with TE.

Reviewer #3:

The manuscript by Dong and colleagues evaluates the potential of human pluripotent stem cells to give rise to trophoblast stem cells. The strengths of the study include: comprehensive evaluation of transcriptional (qPCR, RNA-seq), chromatin (ATAC-seq), and functional (self-renewal, and bipotent differentiation) readouts of the trophoblast stem cell phenotype, systematic comparison of trophoblast differentiation potential between naive and primed pluripotency starting cell states, clear and convincing presentation, and highly significant work.

The main weakness of the manuscript is that the authors did not take the opportunity to directly compare pluripotent stem cell-derived trophoblast stem cell lines with human embryo-derived stem cell lines (derived by Okae et al., 2018) in the majority of their assays (including self-renewal and differentiation potential). The authors did, however, compare chromatin states between pluripotent stem cell-derived trophoblast stem cells and embryo-derived trophoblast stem cells, which revealed numerous differences. They also compared their own RNA-seq data analyses with those published by Okae et al., which is an excellent start, but not as fleshed out as it could be. For these reasons, it is probably premature to state, as the title does, that the authors have, indeed, derived "trophoblast stem cells from naïve human pluripotent stem cells."

Nevertheless, the authors present an abundance of compelling data that they have achieved something new and important.

For these reasons, I am less concerned with whether embryo-derived and stem cell-derived trophoblast stem cell lines are identical, which leads my evaluation to focus instead on addressing the question of whether the authors' major conclusions are supported by the data presented. In the Discussion, the authors state the manuscript's major conclusions:

1) Conclusion 1: Naïve, but not primed, pluripotent stem cell lines give rise to self-renewing trophoblast stem cell lines capable of undergoing differentiation to specialized cell types.

I agree with this statement with a couple of caveats. A) The authors have not shown that pluripotent stem cell-derived trophoblast stem cell lines are equivalent to bona fide, embryo-derived trophoblast stem cell lines, especially given the report that human embryo-derived trophoblast stem cell lines do not express CDX2 (Okae et al., 2018), while the authors putative trophoblast stem cell lines do (Figure 2D). The authors should address this difference and label their new cell line more carefully throughout the manuscript. B) The authors have shown that the population of apparent trophoblast stem cells can give rise to specialized cell types, but I am sure the authors would agree that they do not know whether the cells are bipotent (as stated in the subsection “hTSCs derived from naïve hPSCs have bipotent differentiation potential”) on an individual level, as they have not performed differentiation assays clonally. This distinction should be stated explicitly.

2) Conclusion 2: Naïve pluripotent stem cell-derived trophoblast stem cell lines share transcriptional similarities with preimplantation trophectoderm.

This may be true, but the transcriptional similarities are not profound (r = 0.52). How would mouse trophoblast stem cells and mouse trophectoderm transcriptomes stack up in this analysis? Additionally, I would caution the authors to consider whether a comparison between trophectoderm and epiblast, as they have performed in Figure 4E, would result in genes that are truly trophectoderm-specific. Since the authors have excluded the inner cell mass cell type called primitive endoderm from this analysis, they are not able to know to what extent the trophectoderm-specific subset is, in fact, trophectoderm-specific. Additional discussion of the nature of the genes/pathways that exhibit similar expression dynamics between embryo and ES-derived trophoblast, and their possible relevance to trophoblast biology, could help the authors make a more persuasive case.

3) Conclusion 3: Naïve pluripotent stem cell-derived trophoblast stem cell lines share chromatin similarities with embryo-derived trophoblast stem cell lines.

Again, the authors reported differences too, and it is not yet clear to me which/how many differences are important/unimportant, nor is it clear exactly which aspects of the ATAC-seq data are being correlated in Figure 4G.

I believe that the authors could address these concerns by using more balanced language, and additional computational analyses, rather than performing additional wet-lab experimentation.

---

## [Author Response]

Essential revisions:The majority of the suggestions raised by all three reviewers seem fairly straightforward to address with bioinformatic analyses and changes to the text. In addition, there are several wet-lab experiments that are aimed at firming up the conclusions and digging a bit deeper into what new things can the cell model tell us. We enclose the full reviews for details.An important point is to establish the final identity of the cells. Human TSCs derived from human blastocysts or placentas following the protocol of Okae et al. acquire a post-implantation character. However, here the authors report that naïve human ESCs treated with Okae's media acquire a pre-implantation character. This conclusion should be supported by comparing the gene expression profile of the authors naïve human TSCs to (1) the canonical human TSCs and (2) to this of the pre-implantation human embryos.

We agree with the reviewer’s comment and performed the requested experiments and analyses:

1) We obtained blastocyst-derived, bona fide BT5 hTSCs generated by Okae et al. and performed RNA-seq on these cells. These primary hTSCs clustered together very closely with our naïve hTSCs in the PCA analysis (Figure 4A). In addition, naïve hTSCs displayed similar or elevated expression of key trophoblast marker genes as compared to primary hTSCs (Figure 4—figure supplement 1A). High similarity between naïve and BT5 hTSCs had also been shown in regard to chromatin accessibility landscape (Figure 4F). We conclude that naïve hTSCs have a very similar gene expression profile compared to primary, blastocyst-derived hTSCs.

2) We also compared our naïve hTSCs to the Zhou et al. scRNA-seq dataset, which contains both pre- and post-implantation human TE, EPI, and PE samples. We found that naïve and primary hTSCs cluster the most closely amongst themselves, as well as with day 10 and day 12 TE (Figure 4E). The hTSCs do not share many common transcriptomic characteristics with the EPI or PE lineages (Figure 4E). This analysis revealed the most significant correlation between naïve hTSCs and day 12 TE (Figure 4E and Figure 4—figure supplement 1B), which has a post-implantation identity.

We have modified our manuscript in light of these new data to reflect the conclusion that naïve hTSCs adopt a bona fide hTSC fate as reported by Okaeet al., and possess a post-implantation identity.

Reviewer #1:The manuscript is very well written, the data is clearly presented, the conclusions are well supported by the experimental data and the findings will be very relevant to the stem cell community. I consider this paper suitable for publication in eLife. I only have a few comments:- A recent manuscript (Mischler et al., 2019) has reported the formation of human TSCs from primed ESCs. The authors should comment on these findings in relation to their findings. Mischler et al. report that the neuroectodermal differentiation observed when primed human ESCs are used as a starting point is abolished if SP1 is added to the medium. This means that given the appropriate conditions primed human ESCs are also competent for TSC specification.

We cited the manuscript and added our interpretation and comments about this finding in the second paragraph of the Discussion. We believe it will be important to clarify the developmental identity of primed hPSC-derived cytotrophoblast (CTB)-like cells (as reported by Mischler et al., 2019). It will also be important to compare the quality of hTSCs derived from these distinct sources, and define transcriptional and/or epigenomic similarities between the various cell types that are capable of differentiation into hTSCs. Such a comparative analysis may reveal the precise molecular characteristics that impart competence for hTSC derivation.

- The most problematic aspect of this manuscript is the final identity of the cells. Human TSCs derived from human blastocysts or human placentas following the protocol of Okae et al. acquire a cytotrophoblast (post-implantation) character. However, here the authors report that naïve human ESCs treated with Okae's media acquire a TE (pre-implantation) character. To support this conclusion they should compare the gene expression profile of their naïve human TSCs and the canonical human TSCs. Moreover, although they compare the naïve human TSCs to pre-implantation human embryos they are missing the comparison between naïve human TSCs and post-implantation human embryos (Zhou et al., 2019). Are the naïve human TSCs more similar to pre-implantation TE than post-implantation TE?

We agree with the reviewer’s comment and performed the requested experiments and analyses. As we mentioned above in response to Essential revisions, we have modified our manuscript in light of these new data to reflect the conclusion that naïve hTSCs are highly similar to bona fide hTSCs as reported by Okaeet al., and possess a post-implantation identity that most closely resembles day 12 TE according to the scRNA-seq analysis performed by Zhou et al.

- The authors report that naïve human ESCs are not readily competent to respond to the BMP4 treatment, but they can respond to the human TSC medium without previous capacitation. This is a very interesting difference and the authors could comment on it further in the Discussion.

We thank the reviewer for this suggestion, and commented on this observation in the third paragraph of the Discussion: “It has been proposed that putative CTB progenitors derived from primed hPSCs following BMP4 treatment may actually correspond to a mesodermal phenotype (possibly extraembryonic mesoderm) (Bernardo et al., 2011). […] Our data indicate that naïve hPSCs are unresponsive to BMP4 treatment (Figure 1C), and may in fact suggest that the BMP4-mediated differentiation protocol is primed-state specific.” This is consistent with recent findings from the Smith laboratory that naïve hPSCs require exit from the naïve state before acquiring the capacity to respond to embryonic lineage cues (Rostovskaya et al., 2019).

- At the end of the Introduction the authors state that they have characterized the human TSCs functionally but no functional experiments (e.g. human-mouse chimeras) are presented. The authors should remove this claim or perform the appropriate experiments.

We agree and have removed this claim.

Reviewer #2:This study demonstrates for the first time that naïve-state human pluripotent stem cells (PSCs) can generate trophoblast stem cells (TSCs) in culture. The work is important because it provides a compelling demonstration that functional differences exist between naïve and primed states of human pluripotency with naïve cells uniquely having the developmental capability to form TSCs. In addition, this relatively straightforward route to producing TSCs will make it much easier to study human trophoblast and other placental cell types. Overall, the study has been well carried out with appropriate methods and the use of multiple cell lines. The data are clear and mostly convincing. Current weaknesses of the manuscript are that the work stops short of using the TSCs for anything particularly new and interesting, and there are individual areas where the results need to be better supported.1) The manuscript would clearly be strengthened if the cell model (PSC to TSC) was used to uncover new aspects of stem cell or trophoblast biology. One possibility would be to test at what point along the naïve to primed capacitation (or re-priming) process do the cells lose their capability to derive proliferative TSCs. This would tell us useful information about the timing of lineage restriction events, and this should be relatively straightforward to test. Another possibility is to focus on the role of TEAD4 in TSC specification that the authors allude to at the end of the manuscript. Along these lines, experiments aimed at uncovering the role of TEAD4 through the identification of target sites or using available small molecular inhibitors to test the requirement for TEAD4 function in PSC to TSC conversion could be suitable.

We thank the reviewer for these suggestions, and performed several experiments to begin to explore the role of YAP-TEAD4 in hTSC specification. Verteporfin, a YAP inhibitor that disrupts YAP-TEAD4 interactions, was recently shown to prevent cavity formation in mouse extended pluripotent stem (EPS) cell derived blastoids (EPS-blastoids) (Li et al., 2019). Therefore, we asked whether Verteporfin also affects the derivation of hTSCs from naïve hPSCs. Verteporfin treatment induced widespread cell death within several days when applied during hTSC derivation (Figure 4—figure supplement 2D). We then tested the effect of Verteporfin on the maintenance of naïve hPSCs, and found that it had a similar detrimental effect on viability. This suggests that YAP-TEAD4 signaling is important for the maintenance of naïve hPSCs and possibly also during their transition into a trophoblast fate. These findings may point to species-specific differences in requirements for YAP-TEAD4 signaling between mouse and human. Unlike in the mouse embryo, where YAP is nuclear in the TE and cytoplasmic in the EPI compartment (Nishioka et al., 2009), YAP is nuclear in both the TE and EPI in human blastocysts (Qin et al., 2016). We confirmed by immunofluorescence analysis that YAP shows nuclear localization in both naïve hPSCs and hTSCs (Figure 4—figure supplement 2E). Thus, YAP-TEAD4 may contribute to both EPI and TE development in human, potentially by targeting different regulatory elements and transcription factors. In support of this interpretation, we detected ATAC-seq peaks containing TEAD4 transcription factor binding motifs at key pluripotency regulators in naive hESCs, including *KLF4, NANOG*, and *DPPA2/4* (Figure 4C; Supplementary file 5). Meanwhile, naïve hTSC-specific DARs containing TEAD4 binding motifs are enriched at loci encoding key trophoblast marker genes such as *ITGA2, KRT7*, and *EGFR* (Supplementary file 6), suggesting that TEAD4 is involved in hTSC specification. This divergence is not unprecedented: TFAP2C, which is a trophoblast-specific transcription factor in mouse, was shown to play an essential role in naïve human pluripotency as well (Pastor et al., 2018). How HIPPO/YAP/TEAD4 signaling differentially regulates EPI versus TE development in human, and how that differs from the mouse paradigm, will be of interest for future studies.

With respect to the reviewer’s other suggestion, we agree that it will be of interest to define the exact time window when competence for hTSC derivation is acquired and lost during the interconversion between naïve and primed states, and have incorporated a statement to this effect in the second paragraph of the Discussion. In this regard, we also discuss emerging reports that both EPS cells (Gao et al., 2019), and CTB-like cells generated from primed cells (by Mischler et al., 2019) seem to have the potential to give rise to hTSC-like cells. By defining transcriptional and/or epigenomic characteristics shared between naïve hPSCs and these cell types, we may be able to pinpoint what confers competence for hTSC derivation at the molecular level.

2) In the Introduction, the authors mention that naïve PSCs maintained in 5i/L/A and t2i/L/Gö have transcriptional and epigenetic features of the human preimplantation epiblast. There are also some subtle differences between naïve PSCs cultured in the two conditions, particularly related to the exact stage of human development that show the closest correspondence. Because of this, I think it is important that the authors test whether naïve PSCs maintained in t2i/L/Gö can also generate proliferative TSCs.

In agreement, we have cultured naïve hPSCs in PXGL, the most recent media formulation for naïve hPSC culture from the Smith laboratory (Bredenkamp et al., 2019b), for 8 passages, then cultured them in hTSC medium. The PXGL-derived naïve hTSCs resemble blastocyst-derived and 5i/LA-derived hTSCs morphologically (Figure 2—figure supplement 1G), are double positive for hTSC surface marker ITGA6 and EGFR (Figure 2—figure supplement 1H), express trophoblast marker gene mRNA at approximately the same level as naïve hTSCs derived from 5i/LA (Figure 2—figure supplement 1I), and show positive immunostaining for trophoblast markers (Figure 2—figure supplement 1J). These data suggest that naïve hTSC-derivation is likely an intrinsic property of naïve hPSCs, regardless of the specific culture conditions used.

3) Although unlikely, it is formally possible that naive PSC cultures contain a mixed population with a small number of pre-existing trophoblast progenitor cells. Can the authors provide evidence to support or refute this possibility? Expanding single-cell naïve PSC clones and demonstrating the capability of these clonally-derived lines to form TSCs would be sufficient.

We thank the reviewer for this thoughtful suggestion. We picked and expanded 3 single-cell naïve hPSC clones, A1, A2, and A3 (Figure 2—figure supplement 2A). We confirmed their naïve molecular characteristics and their lack of hTSC marker expression (Figure 2—figure supplement 2B, C). Subsequently, naïve hTSCs were derived from these clonally expanded naïve hPSCs. All of these clonally derived naïve hTSCs exhibit a typical hTSC morphology (Figure 2—figure supplement 2A), and are negative for naïve hPSC markers but positive for hTSC markers (Figure 2—figure supplement 2B, C). The results lend further support to the notion that the ability to give rise to hTSCs is an intrinsic property of naïve hPSCs, rather than representing the expansion of a small population of pre-existing trophoblast-like cells in the culture.

4) It would be useful to see how the naïve TSCs compare to the blastocyst-derived TSCs. The authors should add the published RNA-Seq data sets from Okae et al. into their PCA plot in Figure 4A.

We agree with the reviewer’s comment and have now performed RNA-seq analysis on the blastocyst-derived BT5 line from Okae et al., 2018. As we mentioned above under Essential Revisions, naïve hTSCs and primary hTSCs are closely clustered together in the PCA analysis (Figure 4A). In addition, naïve hTSCs displayed comparable or elevated expression of key trophoblast marker genes as primary hTSCs (Figure 4—figure supplement 1A). High similarity between naïve and BT5 hTSCs had also been shown in regard to chromatin accessibility landscape (Figure 4F). We conclude that naïve hTSCs have a very similar gene expression and chromatin accessibility profile compared to primary, blastocyst-derived hTSCs.

5) Figure 2E: images are poor quality and it is difficult to tell if there is much signal in the cells.

We thank the reviewer for pointing this out. We have now used better quality images in Figure 2E.

6) Related to Figure 4I, if you call differentially accessible regions between blastocyst-derived TSCs versus naive PSCs and also between naive TSCs versus naive PSCs, do you get the same TF motifs? (My understanding is that current Figure 4I data show merged data sets).

We thank the reviewer for the comment. Now we have performed transcription factor binding motif analysis on differentially accessible regions specific to naïve hESCs vs. naïve hTSCs as well as naïve hESCs vs. BT5 hTSCs (Figure 4G). Highly similar motif enrichments were observed for naïve hTSCs and BT5 hTSCs (Figure 4G), which further corroborates their common identity.

7) I think some of the text could be better balanced to help interpret the results and to communicate how the various cell types might correspond to embryo development. The authors start out by showing primed PSCs undergo differentiation in response to BMP signalling, whereas naïve PSCs do not. Based on a set of markers, and in line with previous studies, the authors conclude that primed PSCs retain some capacity to form trophoblast. This has never made much sense from a developmental point of view and in some ways detracts from the more novel finding here that naïve PSCs really can make trophoblast. Whether the BMP4-treated primed PSCs are trophoblast is a matter of debate with at least one paper suggesting that they are more similar to extraembryonic mesoderm, which share many of the marker genes to trophoblast (PMID: 21816365). The other point is related to the comparisons between TSCs and preimplantation trophectoderm (TE) later in the manuscript. This is a reasonable comparison to make, but it is very unlikely that TE contain a highly proliferative trophoblast progenitor, and so I think it is important to communicate that the TSCs are not expected to align too closely with TE.

We thank the reviewer for these helpful comments, and have performed more analysis and modified our text accordingly.

In the third paragraph of the Discussion, we stated that it has been proposed that putative CTB progenitors derived from primed hPSCs following BMP4 treatment may actually correspond to an extraembryonic mesoderm identity (Bernardo et al., 2011). A recent study from the Parast lab confirms that BMP4 treatment of primed hPSCs gives rise to a mixture of trophoblast and mesoderm cells (Horii et al., 2019). Our data indicate that naïve hPSCs are unresponsive to BMP4 treatment (Figure 1C), and may in fact suggest that the BMP4-mediated differentiation protocol is primed-state specific and does not allow for bona fide trophoblast differentiation.

We also compared our naïve hTSCs to the Zhou et al. scRNA-seq dataset, which contains both pre- and post-implantation human TE, EPI, and PE samples. This analysis revealed the most significant correlation between naïve hTSCs and day 12 TE (Figure 4E and Figure 4—figure supplement 1B), which has a post-implantation identity. In light of this, we have revised our interpretation to reflect that hTSCs most likely acquire a post-implantation trophoblast identity.

Reviewer #3:[…] The main weakness of the manuscript is that the authors did not take the opportunity to directly compare pluripotent stem cell-derived trophoblast stem cell lines with human embryo-derived stem cell lines (derived by Okae et al., 2018) in the majority of their assays (including self-renewal and differentiation potential). The authors did, however, compare chromatin states between pluripotent stem cell-derived trophoblast stem cells and embryo-derived trophoblast stem cells, which revealed numerous differences. They also compared their own RNA-seq data analyses with those published by Okae et al., which is an excellent start, but not as fleshed out as it could be. For these reasons, it is probably premature to state, as the title does, that the authors have, indeed, derived "trophoblast stem cells from naïve human pluripotent stem cells."Nevertheless, the authors present an abundance of compelling data that they have achieved something new and important.For these reasons, I am less concerned with whether embryo-derived and stem cell-derived trophoblast stem cell lines are identical, which leads my evaluation to focus instead on addressing the question of whether the authors' major conclusions are supported by the data presented. In the Discussion, the authors state the manuscript's major conclusions:1) Conclusion 1: Naïve, but not primed, pluripotent stem cell lines give rise to self-renewing trophoblast stem cell lines capable of undergoing differentiation to specialized cell types.I agree with this statement with a couple of caveats. A) The authors have not shown that pluripotent stem cell-derived trophoblast stem cell lines are equivalent to bona fide, embryo-derived trophoblast stem cell lines, especially given the report that human embryo-derived trophoblast stem cell lines do not express CDX2 (Okae et al., 2018), while the authors putative trophoblast stem cell lines do (Figure 2D). The authors should address this difference and label their new cell line more carefully throughout the manuscript. B) The authors have shown that the population of apparent trophoblast stem cells can give rise to specialized cell types, but I am sure the authors would agree that they do not know whether the cells are bipotent (as stated in the subsection “hTSCs derived from naïve hPSCs have bipotent differentiation potential”) on an individual level, as they have not performed differentiation assays clonally. This distinction should be stated explicitly.

We thank the reviewer for these suggestions. (A) As shown in Figure 2D and Figure 4—figure supplement 1A, our naïve hTSCs do NOT express CDX2, just as reported for primary hTSCs by Okae et al. This is a further indication that these cells acquire a post-implantation trophoblast identity, since CDX2 is expressed in human pre-implantation TE (Petropoulos et al., 2016; Zhou et al., 2019). (B) We fully agree, and now state this caveat in the subsection “Global transcriptome and chromatin accessibility profiles of hTSCs derived from naïve hPSCs”.

2) Conclusion 2: Naïve pluripotent stem cell-derived trophoblast stem cell lines share transcriptional similarities with preimplantation trophectoderm.This may be true, but the transcriptional similarities are not profound (r = 0.52). How would mouse trophoblast stem cells and mouse trophectoderm transcriptomes stack up in this analysis? Additionally, I would caution the authors to consider whether a comparison between trophectoderm and epiblast, as they have performed in Figure 4E, would result in genes that are truly trophectoderm-specific. Since the authors have excluded the inner cell mass cell type called primitive endoderm from this analysis, they are not able to know to what extent the trophectoderm-specific subset is, in fact, trophectoderm-specific. Additional discussion of the nature of the genes/pathways that exhibit similar expression dynamics between embryo and ES-derived trophoblast, and their possible relevance to trophoblast biology, could help the authors make a more persuasive case.

In agreement, we performed additional analysis to compare our naïve hTSCs to the scRNA-seq dataset from Zhou et al., 2019, which includes pre- and post-implantation TE, EPI, and PE samples. Marker genes of all three lineages as published by Zhou et al. were used for this analysis. We found that naïve hTSCs and primary hTSCs cluster the most closely amongst themselves, as well as with day 10 and day 12 TE (Figure 4E), and thus with post-implantation trophectoderm. The correlation between naïve hTSC/hPSC expression in vitro and TE/EPI expression at day 10 or day 12 in vivo increases to r = 0.61 (Figure 4—figure supplement 1C). We should note that due to many variables, including library preparation methods, sequencing platforms, and bulk vs. single cell RNA-seq differences, the hTSCs in our study are not expected to correlate perfectly with in vivo trophoblast. This analysis also revealed clearly that hTSCs correlate very poorly with either EPI or PE lineages (Figure 4E).

To further uncover the genes and biological processes that contribute to the similarities between naïve hTSCs and post-implantation TE, we first compiled a list of genes that are specifically expressed in either TE or EPI at day 10 or day 12 of human development (Zhou et al., 2019), which are the stages with which naïve hTSCs showed the strongest correlation (Figure 4E). We then performed gene ontology (GO) analysis on the top 10% most highly upregulated genes in naïve hTSCs relative to naïve hPSCs (Figure 4—figure supplement 1C and Supplementary file 4). The most enriched biological processes in naïve hTSCs include female pregnancy, reproductive processes, and positive regulation of NF-kappa B transcription factor activity (Figure 4—figure supplement 1D). This provides additional validation that the process of hTSC derivation from naïve hPSCs activates genes and pathways that are relevant to human trophoblast biology.

3) Conclusion 3: Naïve pluripotent stem cell-derived trophoblast stem cell lines share chromatin similarities with embryo-derived trophoblast stem cell lines.Again, the authors reported differences too, and it is not yet clear to me which/how many differences are important/unimportant, nor is it clear exactly which aspects of the ATAC-seq data are being correlated in Figure 4G.

In Figure 4F (original Figure 4G), all accessible chromatin regions are being correlated, and we rephrased the fifth paragraph of the subsection “Global transcriptome and chromatin accessibility profiles of hTSCs derived from 250 naïve hPSCs” to remove any ambiguity. As shown by the dendrogram in Figure 4F, AN naïve hTSCs are more closely correlated with BT5 hTSCs than with H9 naïve hTSCs, suggesting that naïve- and blastocyst-derived hTSCs have highly similar chromatin accessibility landscapes.

I believe that the authors could address these concerns by using more balanced language, and additional computational analyses, rather than performing additional wet-lab experimentation.